# Transdifferentiation of fibroblasts into muscle cells to constitute cultured meat with tunable intramuscular fat deposition

Tongtong Ma[1†], Ruimin Ren[1,2†], Jianqi Lv[1†], Ruipeng Yang[1], Xinyi Zheng[1], Yang Hu[3], Guiyu Zhu[1], Heng Wang[1]*

[1]College of Animal Science and Technology, Key Laboratory of Efficient Utilization of Non-Grain Feed Resources, Ministry of Agriculture and Rural Affairs, Shandong Agricultural University, Taian, China; [2]College of Animal Science and Technology, Huazhong Agricultural University, Wuhan, China; [3]College of Food Science and Technology, Huazhong Agricultural University, Wuhan, China

*For correspondence:
wangheng@sdau.edu.cn

[†]These authors contributed equally to this work

**Abstract** Current studies on cultured meat mainly focus on the muscle tissue reconstruction in vitro, but lack the formation of intramuscular fat, which is a crucial factor in determining taste, texture, and nutritional contents. Therefore, incorporating fat into cultured meat is of superior value. In this study, we employed the myogenic/lipogenic transdifferentiation of chicken fibroblasts in 3D to produce muscle mass and deposit fat into the same cells without the co-culture or mixture of different cells or fat substances. The immortalized chicken embryonic fibroblasts were implanted into the hydrogel scaffold, and the cell proliferation and myogenic transdifferentiation were conducted in 3D to produce the whole-cut meat mimics. Compared to 2D, cells grown in 3D matrix showed elevated myogenesis and collagen production. We further induced fat deposition in the transdifferentiated muscle cells and the triglyceride content could be manipulated to match and exceed the levels of chicken meat. The gene expression analysis indicated that both lineage-specific and multifunctional signalings could contribute to the generation of muscle/fat matrix. Overall, we were able to precisely modulate muscle, fat, and extracellular matrix contents according to balanced or specialized meat preferences. These findings provide new avenues for customized cultured meat production with desired intramuscular fat contents that can be tailored to meet the diverse demands of consumers.

## eLife assessment

This study presents an **important** new technology for transdifferentiation of fibroblasts into muscle cells. The data and methods used for analysis were **compelling**. This study will have broad interest to cellular reprogramming biologists in particular as well as the general public.

## Introduction

Cultured meat is an innovative and emerging technique that produces meat directly from cell cultures, potentially providing a high-quality, safe, and stable source of animal protein (*Chriki and Hocquette, 2020*). In comparison to traditional livestock and poultry farming, cultured meat generates fewer greenhouse gases, utilizes less arable land, and causes less animal harm (*Mattick et al., 2015*). Proper selection of seed cells is one of the keys to the success of cultured meat production. The starting cells must be easily obtainable and be able to proliferate numerous times to enable mass production. While the stem cells like muscle stem cells and pluripotent stem cells have frequently been utilized

as the cellular source for cultured meat, these stem cells are rare in the animal body and difficult to obtain and amplify on a large scale. In contrast, the somatic cells, which constitute the body, could be efficiently converted into muscle cells under certain conditions. The fibroblast is one of the most abundant and widely distributed cell types present in the body and could be easily collected via minimally invasive biopsy procedure without sacrificing the farm animals. The fibroblasts can replicate indefinitely in vitro and are amenable to myogenesis, adipogenesis, and chondrogenesis (*French et al., 2004*; *Yin et al., 2010*), which produce muscle, fat, and extracellular matrix (ECM) proteins that constitute the meat and the associated texture and flavor. Recently, the fibroblast cells from farm animals have been utilized as the source cells for cultured meat production, with or without myogenesis (*Jeong et al., 2022*; *Pasitka et al., 2023*), demonstrating the feasibility of somatic cell-derived seed cells as a sustainable and ethical option for cultured meat production.

We have previously developed a protocol for the controlled transdifferentiation of chicken fibroblasts into myoblasts, which subsequently form multinucleated myotubes and express mature muscle proteins (*Ren et al., 2022*). Moreover, chicken fibroblasts could also be efficiently converted into fat-depositing lipocytes by treating them with chicken serum (CS) medium or other substances such as fatty acids (*Hausman, 2012*; *Kim et al., 2020*; *Lee et al., 2021*). Therefore, the induced myogenic and adipogenic competency, along with the inherent fibrogenic collagen-producing ability of chicken fibroblasts, can enable us to simultaneously synthesize muscle, fat, and collagen during the cultured meat production. Nevertheless, further techno-functional research is necessary to precisely control the composition of the end product of fibroblast-derived cultured meat in order to achieve a more balanced and personalized nutritional profile and meet the specific consumer preferences.

In this study, the chicken fibroblast cells were implanted into the hydrogel scaffold to analyze the 3D cellular dynamics involving cell proliferation and myogenic/adipogenic transdifferentiation. We also optimized the low-serum culture conditions of chicken fibroblasts to reduce the cost of mass production. The myogenic transdifferentiated cells were confirmed to be skeletal muscle lineage but not myofibroblasts. Importantly, the cells were subjected to myogenesis and adipogenesis sequentially in 3D hydrogel matrix to resemble the whole-cut meat with the controllable intramuscular fat and collagen content. By using transdifferentiation strategies, the depositing ratio of fat into the cultured meat could be manually and precisely adjusted, allowing the nutrients to be naturally synthesized from the organized muscle structure. This demonstrates the potential for manipulating cultured meat to meet consumer preferences for specific fat content and texture.

## Results

### Chicken fibroblasts proliferate stably in low-serum conditions

The chicken fibroblast cells were chosen as the ideal cell source for cultured meat production because they can propagate indefinitely and undergo myogenesis whenever the induction is provided. These cells can be readily obtained from fertilized eggs without the need to harvest animals. We have previously constructed an inducible myogenic transdifferentiation system in chicken fibroblasts with the stable integration of Tet-On-MyoD cassette (*Ren et al., 2022*). The MyoD is the key myogenic transcription factor, and the chicken fibroblasts could be converted into striated and elongated myotubes (myofibers) upon forced expression of MyoD (*Ren et al., 2022*; *Weintraub et al., 1989*). The Tet-On-MyoD cassette enables the inducible and reversible activation of chicken MyoD factor, and in the current fibroblast cells, the myogenic transdifferentiation is only switched on by adding doxycycline (DOX). Without the DOX treatment, the control chicken fibroblast cells (Tet-On-MyoD) do not differ from the wild-type cells in terms of morphology, proliferation rate, and gene expression (*Figure 1—figure supplement 1A–C*). Hence, as a proof-of-concept experiment, we utilized this inducible myogenic fibroblast cell line to develop protocols for cultured chicken meat production.

To achieve sustainable and cost-effective cell production, it is essential to minimize the serum usage in the culture medium. Therefore, we implemented a progressive serum reduction approach to acclimate the cells in the prospect of obtaining a stable cell source that can be propagated in low serum concentrations. In the initial experiment, we used 12% fetal bovine serum (FBS) in 1640 basal medium, which served as the control group. The results showed that the cells were able to proliferate normally at 6, 3, and 1.5% serum concentrations, but at decreasing rates as shown by the EdU assay (*Figure 1A and B*). We also tested the CS as an alternative to bovine serum in order to

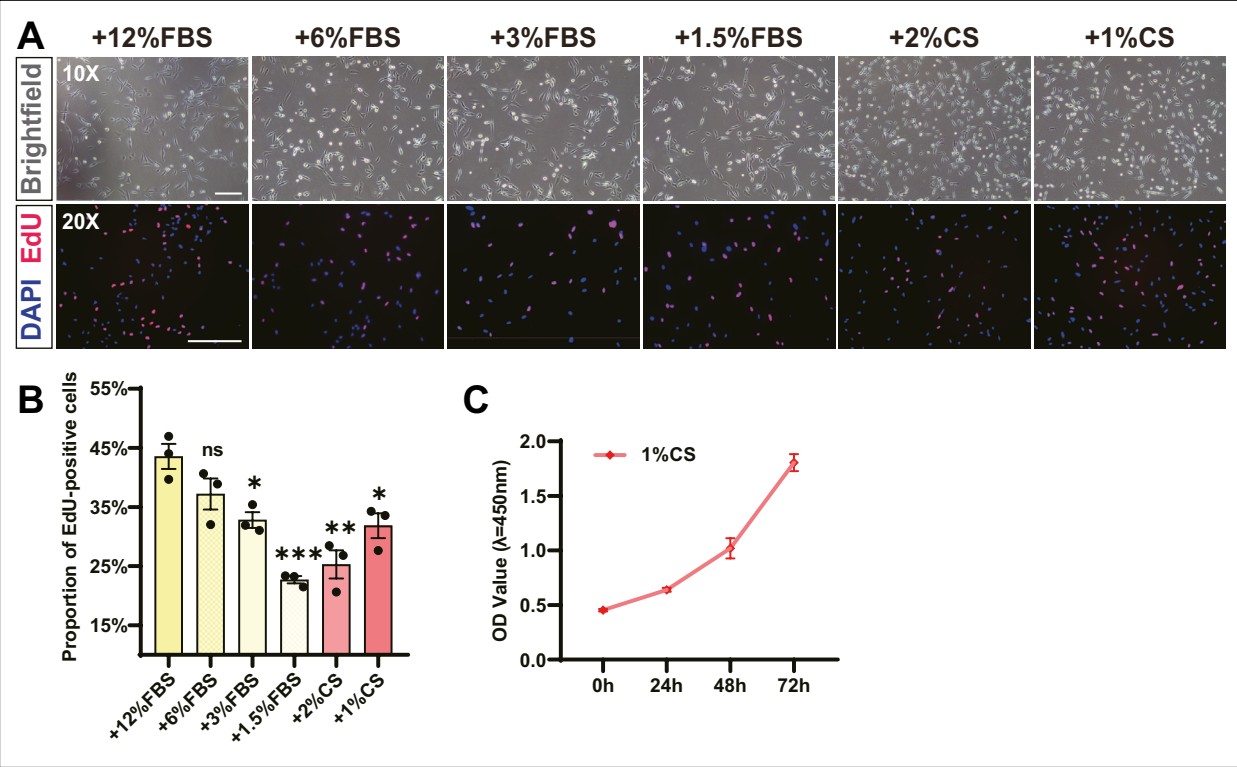

**Figure 1.** Chicken fibroblasts proliferate stably in low-serum conditions. (**A**) Cellular morphology and EdU staining of chicken fibroblasts under different low-serum conditions. FBS: fetal bovine serum; CS: chicken serum. Scale bars, 200 μm. (**B**) Quantification of the proportion of EdU-positive cells in (A). Error bars indicate s.e.m. n = 3. *p<0.05, **p<0.01, ***p<0.001. Paired *t*-test. (**C**) The CCK-8 cell proliferation assay showed the proliferation of chicken fibroblasts in 1% CS. Error bars indicate s.e.m, n = 3.

The online version of this article includes the following figure supplement(s) for figure 1:

**Figure supplement 1.** Experimental scheme of myogenic transdifferentiation.

avoid cross-species contamination of animal derivatives. The results showed that chicken fibroblasts could proliferate stably in the 1% CS (*Figure 1A and B*, *Figure 1—figure supplement 1D*) and the cell populations multiplied three times in 3 d as demonstrated by the CCK-8 assay (*Figure 1C*). Thus, we conclude that the low-serum medium can effectively support the stable propagation of chicken fibroblasts.

## 3D culture of chicken fibroblasts in GelMA hydrogels

The behavior of cells grown on the top of 2D flat surface may differ from that of cells in 3D space. To simulate the 3D natural growth environment of cells in vivo, we utilized the gelatin methacrylate (GelMA)-based hydrogels as scaffolds for chicken fibroblasts. GelMA hydrogel can form a stable and porous structure for cell implantation and is commonly used in tissue engineering because of its great biocompatibility and mechanical tenability (*Pepelanova et al., 2018*). We created hydrogels with varying concentrations at 3, 5, 7, and 9wt%, and then observed their surface characteristics when immersed in culture medium using an emission scanning electron microscope. The porosity was measured to be 83.27, 65.01, 62.47, and 57.96%, respectively. Thus, the higher the mass concentration of hydrogels, the tighter the 3D mesh structure formed and the smaller the porosity (*Figure 2A*). The pore diameters in the scaffolds were determined to range from 3 μm$^2$ to 100 μm$^2$, showing the biggest pore size in 3% hydrogel and the smallest in 9% hydrogel (*Figure 2B*). Therefore, the porosity and pore size of the scaffold could be adjusted to achieve optimal physical strength and nutrients delivery that are suited for long-term cell culturation.

To determine the most ideal gel concentration and pore size suited for chicken fibroblast attachment and growth, we conducted an experiment where we implanted the same number of cells into gels with varied pore sizes and examined the cellular dynamics over a 9-day period (*Figure 2—figure*

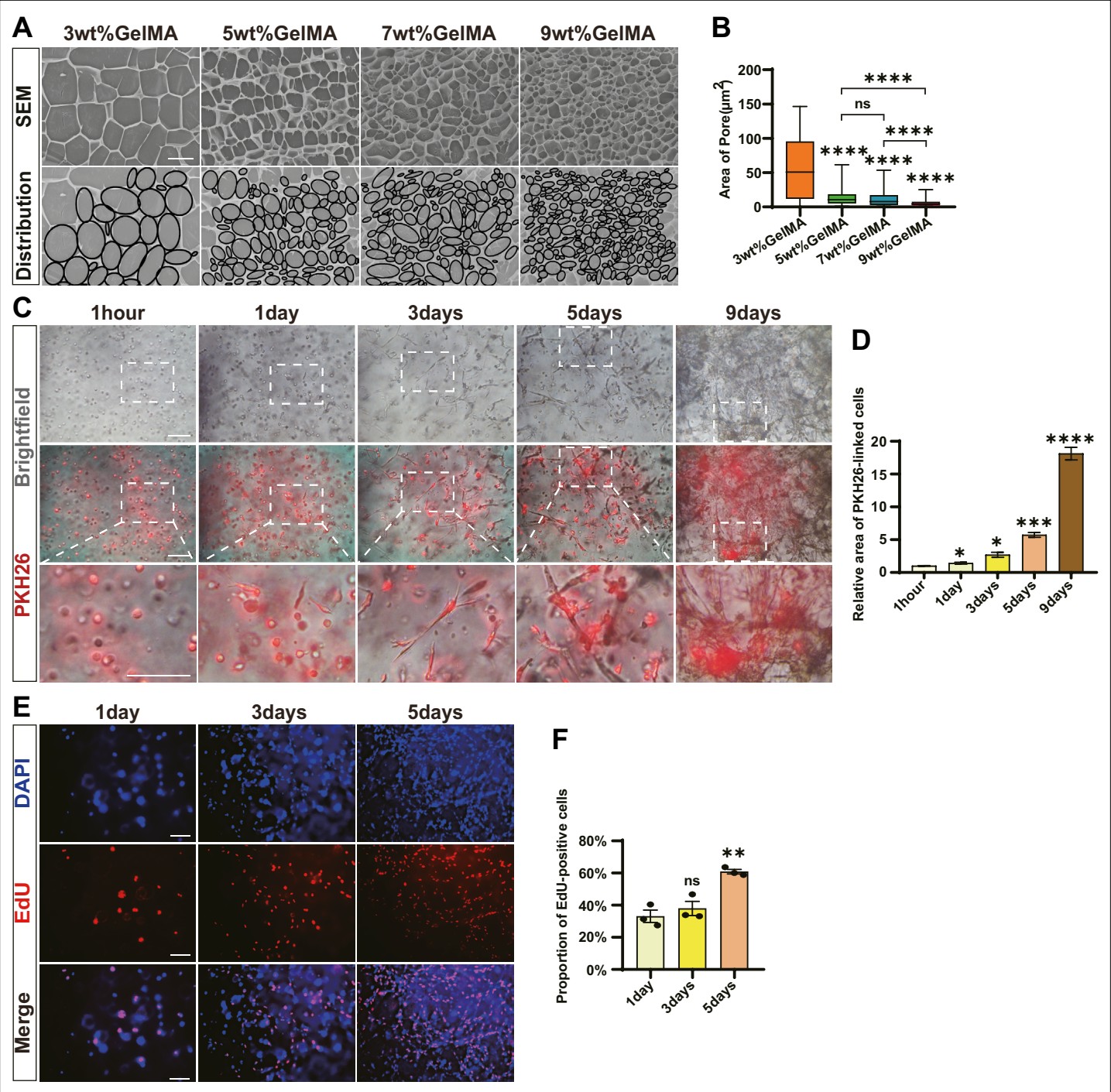

**Figure 2.** 3D culture of chicken fibroblasts in gelatin methacrylate (GelMA) hydrogels. (**A**) Microscopic images of GelMA hydrogels at different concentrations (3, 5, 7, and 9 wt%) taken by scanning electron microscopy (SEM) and their corresponding simplified maps of pore distributions. Scale bar, 10 μm. (**B**) Quantification of pore area in (A). Error bars indicate s.e.m, n = 3. ****p<0.0001. Paired *t*-test. (**C**) Brightfield and red fluorescent images of cells in 3D culture after PKH26 staining at different times (1 hr, 1 d, 3 d, 5 d, and 9 d). Scale bars, 100 μm. (**D**) Relative area of PKH26-linked cells in (C). Error bars indicate s.e.m, n = 3. *p<0.05, ***p<0.001, ****p<0.0001. (**E**) Representative EdU staining shows the proliferation of cells in 3D culture on 1 d, 3 d, and 5 d after cell implantation in hydrogel. Scale bars, 100 μm. (**F**) Quantification of the proportion of EdU-positive cells in (E). Error bars indicate s.e.m, n = 3. **p<0.01. Paired *t*-test.

The online version of this article includes the following figure supplement(s) for figure 2:

**Figure supplement 1.** Cellular morphology of chicken fibroblasts cultured in 3D.

**Figure supplement 2.** The labeling of cells with PKH26 and comparisons of cell morphology and proliferation between 2D and 3D.

*supplement 1*). The cells adhered to the gels immediately and started to exhibit typical extended and irregular fibroblast cell morphology after 1 d, indicating the hydrogel's good biological compatibility with chicken fibroblast cells. However, the 3% hydrogel collapsed in the medium 2 d after cell implantation and thus did not support the long-term cell growth. We continued to monitor the 3D cell proliferation and found that the cells successfully propagated in 5, 7, and 9% hydrogels for the entire duration of the experiment. Among the tested hydrogels, the 5% concentration was found to be the best option for cell attachment and growth as shown by the densest cellular structure formed, and thus be utilized for the subsequent analysis.

Due to the challenges of observing and distinguishing the cells and pores within the hydrogel scaffold using the brightfield of ordinary light microscopy, we then used the PKH26 fluorescent cellular dye to accurately observe the cellular morphology and quantify the cell numbers. The cells were firstly labeled with PKH26 in 2D culture (*Figure 2—figure supplement 2A*) and then transferred into the gel matrix. Upon implantation, the cells appeared mostly round on the first day, but gradually extended and expanded within the hydrogel scaffold. By the fifth day, most of the cells were elongated with irregular shape and short tentacles and packed tightly, and by the ninth day they multiplied more than 15 times and formed dense fibrous bundles (*Figure 2C and D*). We also examined the proliferation dynamics of the cells in 3D scaffold via the EdU assay. The proliferation efficiency of chicken fibroblast cells gradually increased over time in 3D culture and reached replication levels even higher than those in 2D culture (*Figures 1B and 2E,F*). This result indicates that 3D culture conditions provide a more favorable environment for cell growth. Moreover, when the 3D cultured cells were detached from the gel by collagenase dissociation (*Figure 2—figure supplement 2B*) and seeded back in 2D monolayer culture plates, the cells again exhibited similar morphology but slightly increased proliferation capacity compared to the original fibroblasts in 2D conditions (*Figure 2—figure supplement 2C and D*). Thus, the chicken fibroblast cells cultured in 3D maintain their normal physiological characteristics and we continue to stimulate the cellular myogenesis and lipogenesis to prepare them for cultured meat production.

## Transdifferentiation of chicken fibroblasts into muscle cells in 3D

Despite the complexity of the composition of fresh meat and processed products, muscle cells are the major and indispensable component of meat foods. The muscle cell-derived myotubes (myofibers) provide a rich source of proteins and nutrients and constitute the meat texture. To this end, we aimed to transform the chicken fibroblast into muscle cells through the established MyoD overexpression protocol (*Ren et al., 2022*). We first tested and optimized the myogenic transdifferentiation parameters in 2D culture through the DOX-induced MyoD expression (*Figure 3—figure supplement 1*). We observed elongated multinucleated myotubes and abundant expression of myosin heavy chain (MHC) after 3 d of myogenic induction (*Figure 3—figure supplement 2A and B*). The 'serum deprivation' protocol was the classical strategy to stimulate terminal myogenic differentiation in myoblasts of various species including chicken (*Nakashima et al., 2005*). Thus, we examined the myogenic transdifferentiation efficiency (myotube fusion index) in the chicken fibroblast cells overexpressing MyoD in combination with reduced concentrations of bovine or horse serums. However, we found that reducing serum concentrations did not increase the myotube formation but instead caused massive cell death (*Figure 3—figure supplement 2B*). Therefore, it seems that the 'serum deprivation' could not further improve the myogenic transdifferentiation in chicken fibroblast cells and a simple MyoD overexpression strategy is sufficient for efficient production of mature muscle cells.

After demonstrating the feasibility of induced transdifferentiation of chicken fibroblasts into muscle cells in 2D culture, we continued myogenic transdifferentiation in 3D to simulate the construction of concrete whole-cut meat. The cells were inoculated into a hydrogel scaffold and allowed to proliferate for 7 d before inducing transdifferentiation with MyoD overexpression (*Figure 3A*). We used whole-amount MHC immunofluorescence staining to examine the myotube formation directly inside the gel. We identified abundant multinucleate MHC$^+$ myotubes at multiple focal planes within the gel (*Figure 3B*, *Figure 3—video 1*), indicating successful myogenic transdifferentiation of 3D cultured chicken fibroblasts. In contrast, no MHC signal or autofluorescence was detected in the 3D cultured chicken fibroblasts (*Figure 3—figure supplement 2C*) and hydrogel without cells (*Figure 3—figure supplement 2D*). With the assistance of confocal Z-stack analysis, the stacked images showed densely packed MHC$^+$ myotubes from a piece of cellular hydrogel complex at a depth of 68 μm (*Figure 3D*).

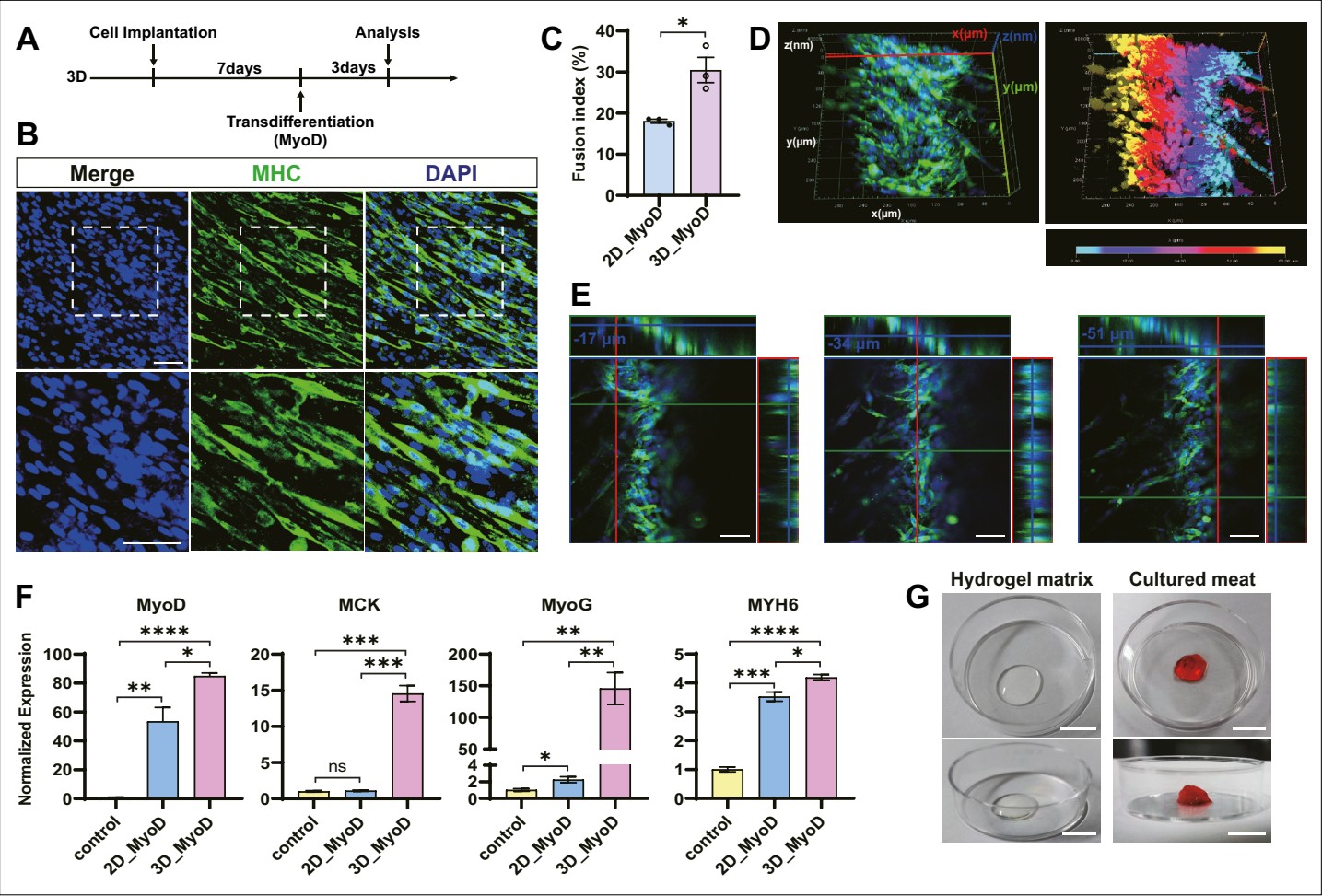

**Figure 3.** Transdifferentiation of chicken fibroblasts into muscle cells in 3D. (**A**) Experimental design for fibroblast myogenic transdifferentiation in 3D culture. (**B**) Representative images of myosin heavy chain (MHC) staining showed the myogenic ability of chicken fibroblasts in 3D culture. Scale bars, 50 μm. (**C**) Comparison of the mean myogenic fusion index between 2D and 3D. Error bars indicate s.e.m, n = 3. *p<0.05. Paired *t*-test. (**D**) 3D images of MHC staining of cells cultured in 3D. The right panel is depth-coded image, which indicate different depths from the deepest (cyan) to the surface (yellow). (**E**) Orthogonal projections of three sets of MHC staining of cells in 3D culture at different depths. Scale bars, 50 μm. (**F**) Expression of skeletal muscle-related genes was determined by RT-qPCR in 2D and 3D cells upon myogenic transdifferentiation and control 3D cells without stimulation. Note that the myogenic transdifferentiation driven by MyoD stimulates the expression of classical myogenic factors. Error bars indicate s.e.m, n = 3. *p<0.05, **p<0.01, ***p<0.001, ****p<0.0001. Paired *t*-test. (**G**) Macroscopic morphology of the empty hydrogel matrix (left) and cultured meat (right). The cultured meat is the product obtained after 3D cell culture and induction of myogenesis. Scale bars, 1 cm.

The online version of this article includes the following video and figure supplement(s) for figure 3:

**Figure supplement 1.** Expression of MyoD upon doxycycline (DOX) treatment.

**Figure supplement 2.** Myogenic transdifferentiation in 2D.

**Figure supplement 3.** Cell gelatin methacrylate (GelMA) 3D culture units and macroscopic morphology after 7 d of culture, with a white plastic frame as the fixation ring.

**Figure 3—video 1.** MHC+ myotubes in the 3D cultured fibroblast after myogenic transdifferentiation.
https://elifesciences.org/articles/93220/figures#fig3video1

**Figure 3—video 2.** Multiangle video showing myotubes were aligned together.
https://elifesciences.org/articles/93220/figures#fig3video2

The separate XY axis views of the orthogonal projections at different depths (*Figure 3E*) and a multiangle video (*Figure 3—video 2*) also showed that several myotubes were aligned together. Nevertheless, many myotubes were oriented in different directions, preventing the entire matrix from aligning in one direction. In conclusion, we successfully transformed the chicken fibroblast cells into mature muscle cells in 3D environment.

Compared to 2D, the transdifferentiated myotubes induced in 3D were more organized and densely packed, resembling the native myofiber distribution in vivo (*Figure 3B*). The myotube formation efficiency (fusion index) in 3D reached 30.49%, which was significantly higher than that of the 2D under the same transdifferentiation conditions (*Figure 3C*). We also evaluated the expression of several myogenic markers by RT-qPCR (*Figure 3F*). MyoG (myogenin) is a transcription factor regulating terminal skeletal muscle differentiation and could be induced by MyoD (*Cao et al., 2006*). MYH6 (myosin heavy chain 6) is the major protein comprising the muscle thick filament and functions in muscle contraction (*van Rooij et al., 2009*). MCK (muscle creatine kinase) is an enzyme that is primarily active in mature skeletal and heart muscle (*Vincent et al., 1993*). All myogenic factors, including MyoG, MYH6, and MCK, significantly increased upon transdifferentiation in both 2D and 3D. In addition, the transdifferentiated cells exhibited significantly higher expression of MyoG and MCK in 3D conditions compared to that in 2D, indicating more robust myogenic differentiation and maturation of cells in the 3D microenvironment. We speculate that the porous structure of the hydrogel matrix may support the cells to grow in all directions, similar to the environment in which the myofibers form in vivo.

Macroscopically, the muscle filaments and the dense cellular network structures formed by myogenic transdifferentiation could make the hydrogel matrix more compact and solid (*Figure 3G*). Thus, compared to the empty transparent scaffolds without cells, the hydrogels cultivated with muscle cells more closely resemble whole-cut meat, similar to fresh animal meat.

## Myogenic transdifferentiation of fibroblast does not produce myofibroblasts

Fibroblasts can be induced to differentiate into myofibroblast upon injury, leading to tissue fibrosis. Similar to the skeletal muscle cells, the myofibroblast is also contractile and expresses the myogenic factor MyoD as well as certain types of sarcomeric MHCs (*Hecker et al., 2011*). To confirm that the myogenic transdifferentiated cells were indeed skeletal muscle cells but not the myofibroblasts, we utilized a panel of cell lineage-specific markers to delineate the cell conversion progress and determine the cellular identity of fibroblasts, myoblasts, and myofibroblasts, respectively. The immunofluorescence staining of 3D cultured cells showed that the muscle-specific intermediate filament Desmin (*Paulin and Li, 2004*) was expressed only in the MyoD-transdifferentiated cells, but not in the original fibroblasts or the embryonic skin-derived myofibroblasts (*Figure 4A*). In contrast, the classical myofibroblast marker alpha-smooth muscle actin (α-SMA) (*Hinz et al., 2007*) was expressed only in myofibroblasts, but not in the fibroblast or the transdifferentiated cells (*Figure 4*). The fibroblast intermediate filament Vimentin (*Tarbit et al., 2019*) was abundantly expressed in the fibroblasts but reduced in the myogenic transdifferentiated cells (*Figure 4C*). The 2D and 3D cultured cells showed consistent pattern of marker protein expression, indicating that the different culture models and conditions do not affect the cell identity conversion (*Figure 4—figure supplement 1A and B*). These results confirmed that the MyoD indeed transdifferentiate the cells toward the skeletal muscle lineage but not the myofibroblast. Furthermore, the RT-qPCR showed that, after myogenic induction in 3D, the skeletal muscle-specific genes Desmin and Six1 (*Relaix et al., 2013*) were significantly elevated (*Figure 4D*), whereas the fibroblast gene Thy-1 was significantly reduced (*Figure 4E*). The transforming growth factor β (TGFβ) signaling is the most potent known inducer of myofibroblast differentiation (*Vaughan et al., 2000*). We found that the expression of core TGFβ signaling components, TGFβ-1, TGFβ-3, and Smad3, remained unchanged during the transdifferentiation process (*Figure 4F*), indicating that the classical myofibroblast lineage was not induced. Together, these data confirm that the myogenic transdifferentiation of fibroblast does not produce myofibroblasts.

## Stimulate the fat deposition in chicken fibroblasts in 3D

The intramuscular fat is a crucial component of meat that can determine its quality attributes, such as taste and flavor. The chicken fibroblasts have been reported to be amenable to lipogenesis through various stimuli, including CS, insulin, fatty acids, and retinoic acids (*Kim et al., 2021*; *Kim et al., 2020*; *Lee et al., 2021*). We first attempted different lipogenic stimulations on 2D cultured cells and stained them for Oil Red O to visualize and quantify fat deposition in the fibroblasts. Very few scattered Oil Red O signals were found in the treatments consisting of only serums (12.5% FBS, 1% CS, or 2% CS), indicating no lipogenesis during normal proliferation conditions. However, when we added insulin

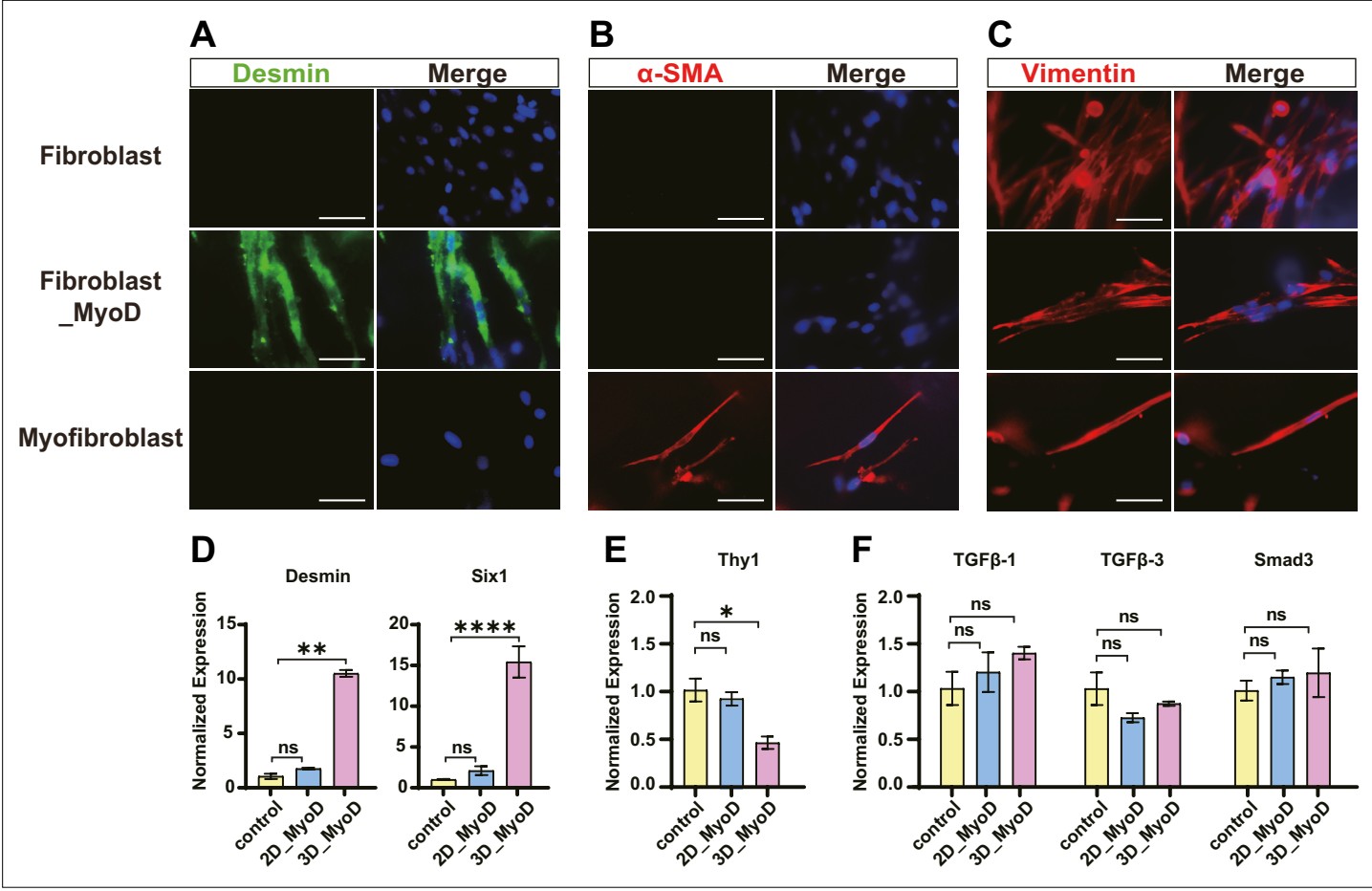

**Figure 4.** Myogenic transdifferentiation of fibroblasts does not produce myofibroblasts. (**A**) Immunofluorescence staining of 3D cultured cells showed that the skeletal muscle marker Desmin was expressed only in the transdifferentiated cells but not in fibroblasts or myofibroblasts. Scale bars, 50 μm. (**B**) Immunofluorescence staining of 3D cultured cells showed that the myofibroblast marker alpha-smooth muscle actin (α-SMA) was expressed only in the myofibroblasts but not in fibroblasts or transdifferentiated cells. Scale bars, 50 μm. (**C**) Immunofluorescence staining of 3D cultured cells showed that the fibroblast marker Vimentin was abundantly expressed in fibroblasts and myofibroblasts but greatly reduced in transdifferentiated cells. Scale bars, 50 μm. (**D**) RT-qPCR showed that the myogenic genes Desmin and Six1 were significantly increased upon myogenic transdifferentiation. (**E**) RT-qPCR showed the fibroblast marker gene Thy-1 was significantly reduced upon myogenic transdifferentiation. (**F**) The myofibroblast marker genes TGFβ-1, TGFβ-3, and Smad3 remain unchanged during myogenic transdifferentiation. Error bars indicate s.e.m, n = 4. *p<0.05, **p<0.01, ***p<0.001, ****p<0.0001. ns: not significant. Paired *t*-test.

The online version of this article includes the following figure supplement(s) for figure 4:

**Figure supplement 1.** Myogenic transdifferentiation of fibroblasts does not produce myofibroblasts in 2D culture.

and fatty acids (oleic/linoleic acid) to the medium, lipid droplets in the cells dramatically increased as detected by the Oil Red O staining. After extensive optimization of the concentrations of supplements in the medium, we identified that the 8 μg/ml fatty acids plus 60 μg/ml insulin (abbreviated as FI, F: fatty acids, I: insulin) can induce lipogenesis most efficiently in 2D chicken fibroblasts (*Figure 5—figure supplement 1A–C*).

Next, the same lipogenesis induction strategy was applied to the 3D cultured cells (*Figure 5A*). We found extensive Oil Red O signal inside the hydrogel matrix at different focal planes and the lipid droplets were clearly visible as beaded strings under magnification (*Figure 5B and C*, *Figure 5—video 1*). The RT-qPCR analysis illustrated the expression of genes involved in lipogenesis and triglyceride synthesis was significantly higher in the cells with lipogenic stimulation compared to the control (*Figure 5D*). Notably, the genes encoding for PPARγ (peroxisome proliferator-activated receptor gamma), Gpd1 (glycerol-3-phosphate dehydrogenase), and FABP4 (fatty acid binding protein 4) all showed higher expression in 3D than that in 2D. Interestingly, the expression of Znf423 (zinc finger protein 423), which is a PPARγ transcriptional activator (*Addison et al., 2014*; *Longo et al., 2018*),

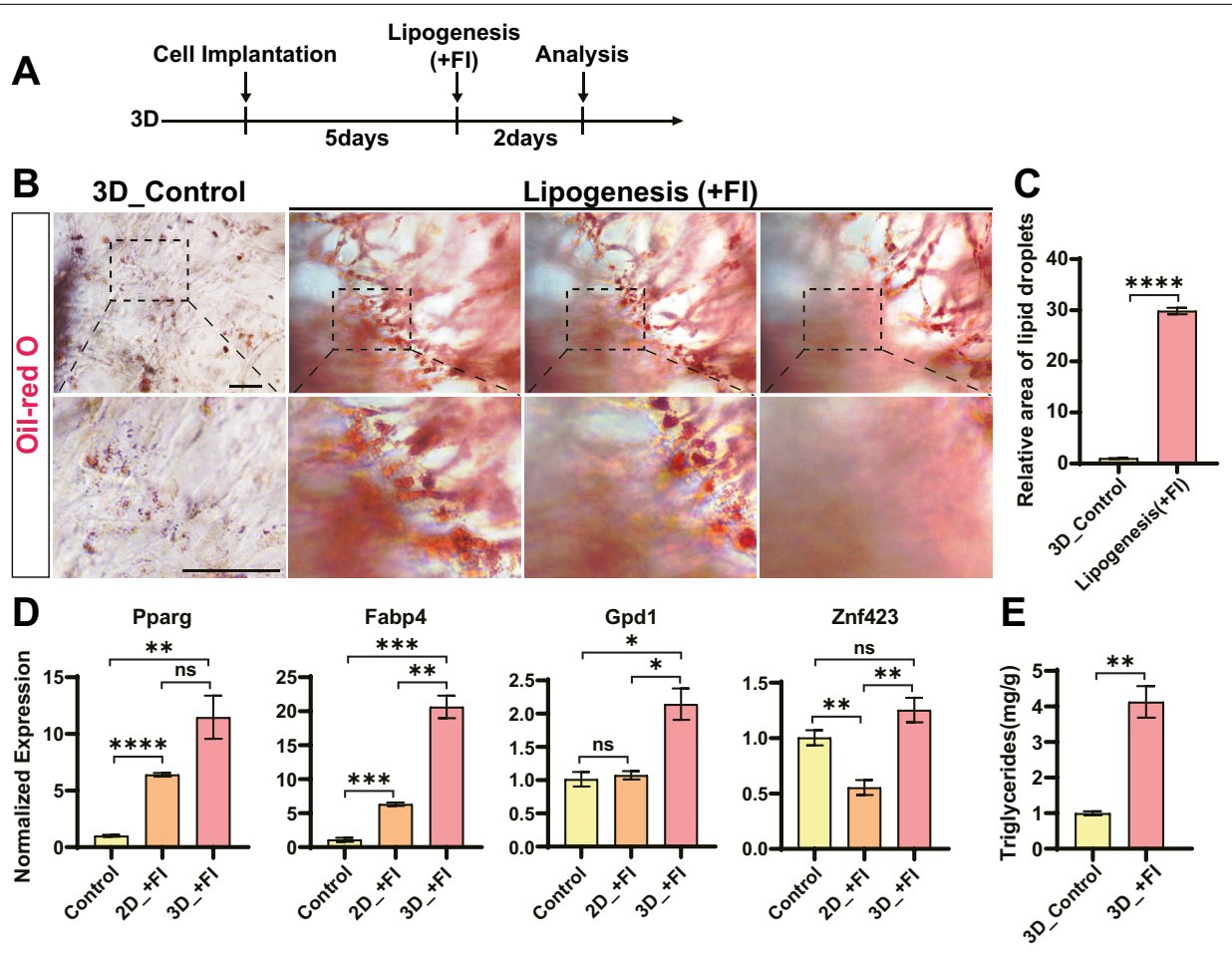

**Figure 5.** Stimulation of fat deposition in chicken fibroblasts in 3D. (**A**) Experimental design for fibroblast lipogenesis in 3D culture ('F' is for fatty acids and 'I' is for insulin). (**B**) Representative images showing the Oil Red O staining of lipid content accumulated in cells at different focal planes at the same position. The control group was normal medium without lipogenesis. Scale bars, 100 μm. (**C**) Relative area of lipid droplets in (B). Error bars indicate s.e.m, n = 3. ****$p<0.0001$. Paired $t$-test. (**D**) Expression of lipid synthesis-related genes determined by RT-qPCR in 2D and 3D cells upon lipogenic induction and control 3D cells without stimulation. Error bars indicate s.e.m, n = 3. *$p<0.05$, **$p<0.01$, ***$p<0.001$, ****$p<0.0001$. Paired $t$-test. (**E**) Triglyceride content in the cultured meat upon different lipogenic inductions and control 3D cells without stimulation. Error bars indicate s.e.m, n = 3. **$p<0.01$. Paired $t$-test.

The online version of this article includes the following video and figure supplement(s) for figure 5:

**Figure supplement 1.** Efficient lipogenesis in 2D chicken fibroblasts.

**Figure supplement 2.** Chicken serum (CS)-induced lipogenesis in 3D cultured chicken fibroblast.

**Figure 5—video 1.** Lipid droplets were observed in 3D cultured fibroblasts upon lipogenic induction.

https://elifesciences.org/articles/93220/figures#fig5video1

only increased upon lipogenic induction in 3D but not in 2D conditions. It seems that the cells grown in 3D hydrogel showed enhanced lipogenesis compared to the flat surface cultured cells, similar to the myogenic transdifferentiation process in 3D. In addition to the lipogenic induction with fatty acids and insulin (FI), we also validated the use of CS in promoting lipid accumulation as previously reported (*Kim et al., 2021*). Our results showed that the CS alone could also stimulate fat accumulation effectively (*Figure 5—figure supplement 2A–C*). We further measured the triglyceride content and found that the lipogenic induction increased the triglyceride significantly in the cell matrix (*Figure 5E*). In conclusion, the 3D cultured chicken fibroblast can efficiently deposit lipids by different stimulations.

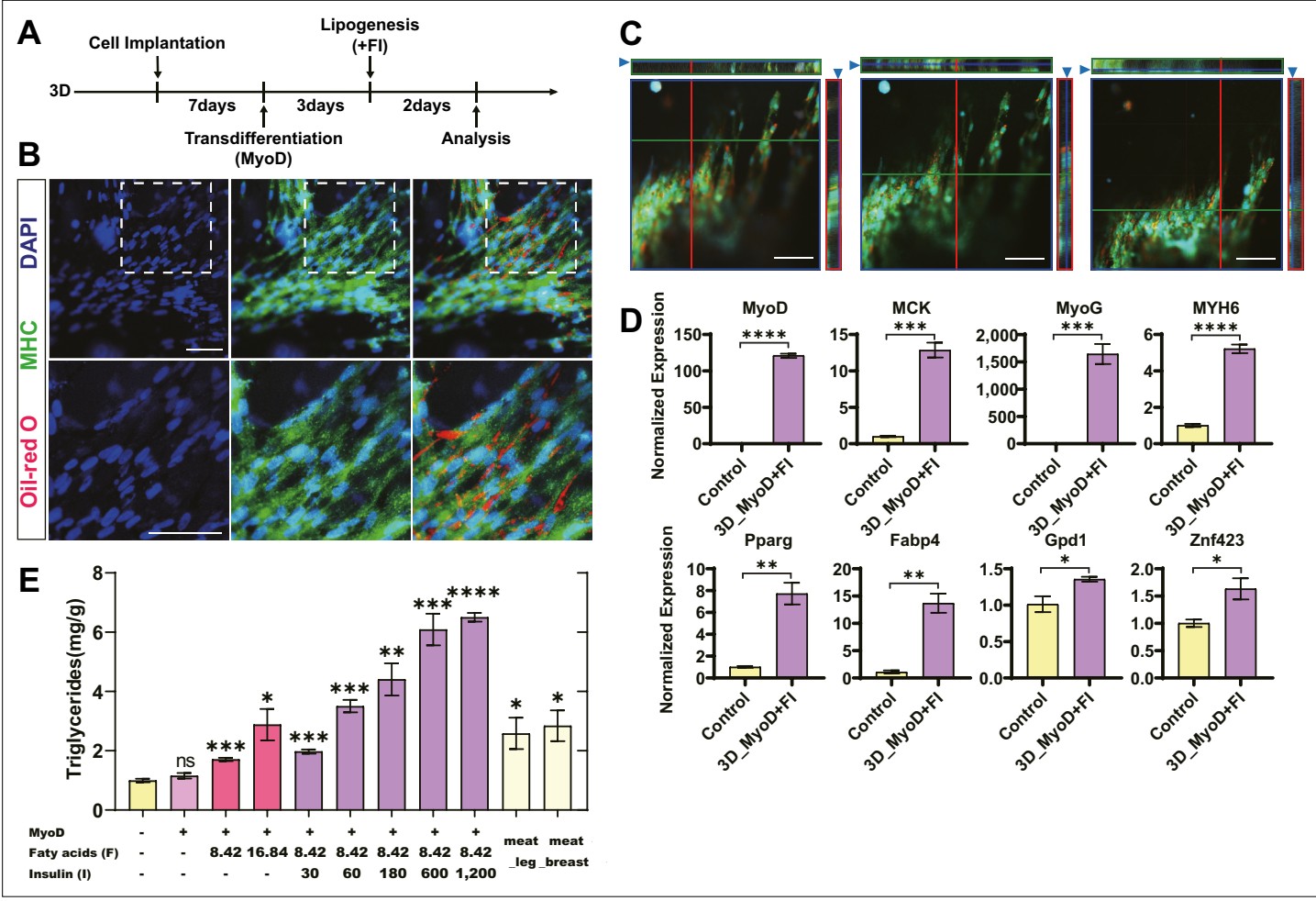

**Figure 6.** Controlled fat deposition in the transdifferentiated muscle cells in 3D hydrogel. (**A**) Experimental design for fibroblast myogenic/lipogenic differentiation in 3D culture. (**B**) Representative images of myosin heavy chain (MHC) and Oil Red O staining of cells upon myogenesis/lipogenesis in 3D culture. Scale bars, 50 μm. (**C**) Orthogonal projections of three sets of MHC and Oil Red O staining of cells in 3D culture at different depths. Scale bars, 50 μm. (**D**) Expression of muscle-related genes (top) and lipid-related genes (bottom) in the cells with myogenesis/lipogenesis induction and control 3D cells without any stimulation were determined by RT-qPCR. (**E**) Triglyceride content of cultured meat under different conditions and real meat compare to fibroblasts_control. 'Meat_leg' and 'Meat_breast' were taken from the leg and breast muscles of adult chickens. Error bars indicate s.e.m, n = 3. *$p<0.05$, **$p<0.01$, ***$p<0.001$, ****$p<0.0001$. Paired $t$-test.

The online version of this article includes the following figure supplement(s) for figure 6:

**Figure supplement 1.** Myogenic/lipogenic stimulation in chicken fibroblasts.

## Controlled fat deposition in the transdifferentiated muscle cells in 3D hydrogel

The above-presented data shows that chicken fibroblast cells have a superior capacity for transforming into muscle and depositing fat when cultured in a 3D hydrogel matrix. Next, we tried to combine the myogenic and lipogenic stimuli together to modulate the fat deposition in the cultured meat to simulate the various intramuscular fat contents in the conventionally raised meat. Rather than converting fibroblasts into muscle cells and fat cells separately and mixing them later, we adopted a new strategy that can induce de novo lipid deposition in the muscle by first inducing myogenic transdifferentiation and then followed by lipid induction in the same cells (*Figure 6A*, *Figure 6—figure supplement 1A*). In 2D conditions, plenty of MHC⁺ myotubes and Oil Red O-stained lipids were found to intermingle after the myogenic/lipogenic treatment (*Figure 6—figure supplement 1B*), and some of the red marked lipid droplets were located inside the myotubes, indicating that the transformed muscle cells indeed deposit fat autonomously to constitute intramyocellular lipids (*Figure 6—figure supplement*

*1C*). Next, we applied similar treatment to the cells cultured in 3D hydrogel and also identified Oil Red O-labeled lipid droplets mixed with the densely packed MHC⁺ myotubes (*Figure 6B and C*). These findings suggest that the use of myogenic/lipogenic treatments can induce the formation of muscle cells that are capable of depositing fat in both 2D and 3D. We further examined the expression levels of both myogenic and lipogenic factors in the 3D cultured cells by RT-qPCR. Compared to the control 3D cells without myogenic/lipogenic stimulations, the induced cells showed significantly higher expression of genes involved in both myogenesis and lipogenesis (*Figure 6D*). Interestingly, the extent of gene upregulation upon the combined myogenic/adipogenic stimulations was comparable to that of myogenic or adipogenic induction alone. This finding suggests that sequential myogenic transdifferentiation and lipid deposition do not interfere with each other when conducted in the same cells.

The intramuscular fat is an integral component of both traditional animal meat and cultured meat, and it directly influences the meat flavor and texture (*Frank et al., 2016*). Hence, we compared the triglyceride levels in the 3D hydrogel cells with different types of lipogenic stimuli with the chicken breast and leg meat. The results showed that the lipogenic stimulation in the 3D hydrogel cells increased the triglyceride content in the cultured meat to the levels comparable to or even higher than real chicken meat (*Figure 6E*). In contrast, the control cells without any induction or with only myogenic stimulation do not show apparent triglyceride accumulation (*Figure 6E*). Therefore, the fat content in the cultured meat could be synthesized in a controlled manner, and then we tried to

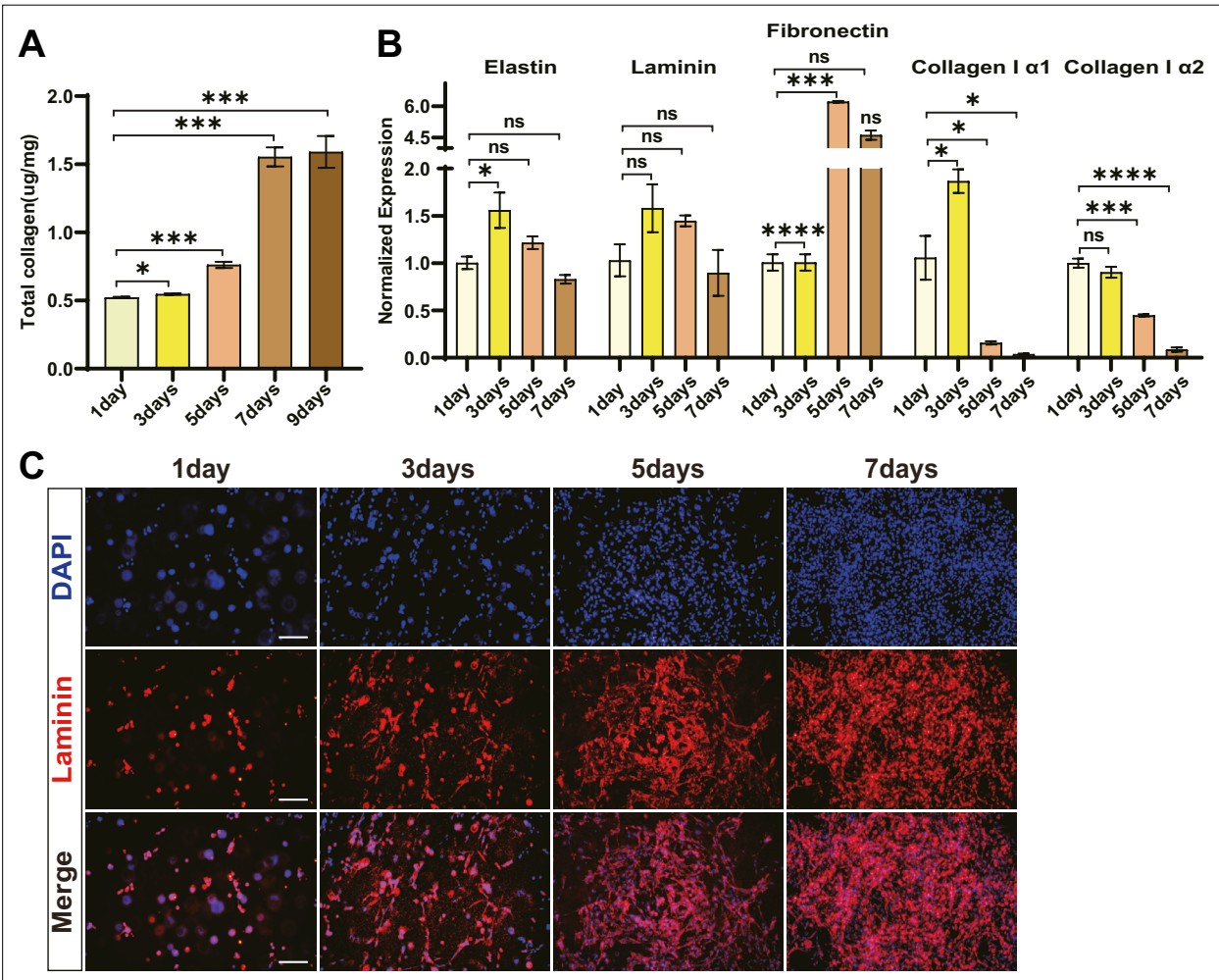

**Figure 7.** The collagen content and expression of extracellular matrix (ECM) components in cultured meat. (**A**) Total collagen content of cultured meat at different days of cultivation. Error bars indicate s.e.m, n = 3. *p<0.05, ***p<0.001. Paired *t*-test. (**B**) Expression of ECM-related genes determined by RT-qPCR of cultured meat. Error bars indicate s.e.m, n = 3. *p<0.05, ***p<0.001, ****p<0.0001. Paired *t*-test. (**C**) Representative Laminin staining of cells in 3D culture on 1 d, 3 d, 5 d, and 7 d after cell implantation and transdifferentiation in hydrogel. Scale bars, 100 μm.

purposely manipulate the triglyceride contents in the meat matrix by adjusting the potency of adipogenic stimulation. By fine-tuning the concentrations of insulin and fatty acids during lipogenic induction, the triglyceride contents in the final product of cultured meat can precisely reach any customized levels across the range from 1.5 mg/g to 7 mg/g (*Figure 6E*), which overlap and surpass the levels in the fresh chicken breast and leg meat. As a result, this strategy greatly expands the diversity and category of cultured meat products, allowing for precise control over intramuscular fat contents to meet consumer preferences. Therefore, guided and graded fat deposition in cultured meat allows for the creation of various meat products with controlled intramuscular fat contents.

## The collagen content and extracellular matrix components of cultured meat

Fibroblasts are an essential source of extracellular matrix (ECM), including the collagen, which provides elasticity to the tissue in the body and enriches the texture of the cultured meat (*Ben-Arye and Levenberg, 2019*). In theory, the fibroblast should generate abundant ECM to produce a more realistic meat product. We then examined the collagen content in the cultured meat and found that the total collagen protein gradually increased and reached the plateau at 1.59 μg/mg in the final product (*Figure 7A*). This is mainly due to the increased cell numbers and the accumulation of secreted collagen in the hydrogel matrix during myogenic/adipogenic transdifferentiation. Nevertheless, the RT-qPCR showed that the genes encoding the major components of ECM exhibited various expression patterns with the extension of culture time. The expression of COL1A1 (collagen, type I, alpha 1) and COL1A2 (collagen, type I, alpha 2) gradually decreased, whereas the fibronectin increased during the time course of meat synthesis. The expression of elastin and laminin genes remained stable throughout the whole course of the experiment (*Figure 7B*). However, the laminin protein content was accumulated and increased steadily during 3D culturation (*Figure 7C*). Overall, the synthesis and accumulation of different types and amounts of ECM components during the myogenic/lipogenic stimulations can improve the texture of the cultured meat prepared from fibroblast cells.

## The characterization of molecular changes during myogenic transdifferentiation and fat deposition in cultured meat

To provide insights into the functional shifts during the transdifferentiation from fibroblasts toward muscle, fat, or muscle/fat cells in 3D culture, we further analyzed the transcriptomes from the different populations of cells including the 'original fibroblasts' (3D_fibroblast), 'myogenic transdifferentiated cells' (3D_MyoD), 'adipogenic transdifferentiated cells' (3D+FI), and 'myogenic/adipogenic transdifferentiated cells' (3D_MyoD+FI) (*Figure 8A*). To illustrate the relationship between these cell groups, we conducted an unsupervised hierarchical clustering analysis of the whole transcriptome. The findings revealed that the 3D+FI group clustered distinctly from the others, while the 3D_MyoD and 3D_MyoD + FI groups exhibited greater similarity. Moreover, the 3D_fibroblasts formed a distinct sub-cluster on their own (*Figure 8B*), suggesting that myogenic or adipogenic transdifferentiation drives these cells away from their original fibroblastic state. The principal component analysis (PCA) of the transcriptomes also showed that distinct trajectories of myogenic and adipogenic transdifferentiation routes were derived from the original fibroblasts and finally integrated together into the myogenic/adipogenic cells (3D) (*Figure 8C*). It indicates that the myogenic and adipogenic signalings could operate simultaneously and separately during the generation of the culture meat composed of muscle and fat. We also compared the differentially expressed genes (DEGs) from '3D_MyoD vs 3D_fibroblast', '3D+FI vs 3D_fibroblast', and '3D_MyoD + FI vs 3D_fibroblast'. The results showed that the majority (78%) of DEGs in the 3D_MyoD+FI are overlapped with 3D_MyoD and 3D+FI, indicating that sequential myogenic/adipogenic induction in 3D_MyoD+FI is consistent with myogenic or adipogenic function individually (*Figure 8D*). The heat map also highlighted the representative myogenic or adipogenic genes that were upregulated in the myogenic, adipogenic, or myogenic/adipogenic cells constitute the culture meat. In contrast, the fibroblast genes were diminished during the transdifferentiation, confirming the loss of fibroblast identity (*Figure 8E*). In addition, the Gene Ontology (GO) analysis of upregulated DEGs in 3D_MyoD+FI cells confirmed that myogenic-specific pathways such as 'muscle organ development' and adipogenic-specific pathways such as 'PPAR signaling pathway' were enriched. In addition, we also identified several multifunctional signaling pathways such as 'JAK-STAT signaling pathway', 'NF-kappa B signaling pathway', and 'MAPK signaling pathway' that

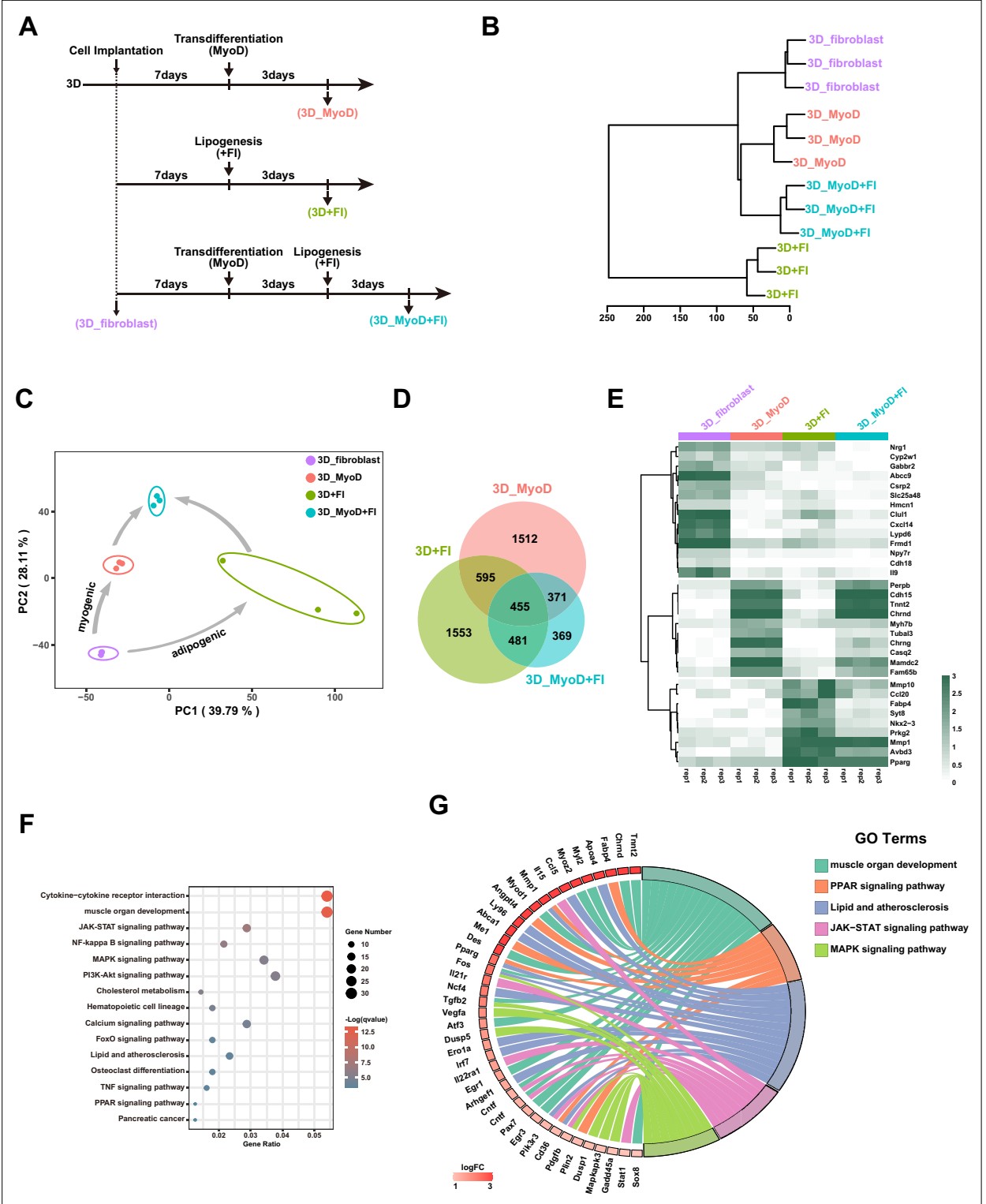

**Figure 8.** Gene expression profiles during transdifferentiation and fat deposition in 3D. (**A**) Scheme of the RNA-seq samples marked by different colors. (**B**) Hierarchical clustering analysis of whole transcriptomes of 3D_fibroblasts, 3D_MyoD, 3D+FI, and 3D_MyoD+FI using Euclidean distance with ward.D cluster method. (**C**) Principal component analysis (PCA) of transcriptome changes during myogenic transdifferentiation and fat deposition (n = 10,247 genes). The ellipses group includes three biological replicates in each cell type. The arrows represent the reprogramming of gene expression under different conditions. The routes were derived from the original fibroblast toward two differentiation routes, namely 'myogenic transdifferentiation' and 'adipogenic transdifferentiation'. (**D**) Venn diagram showing the overlap of differentially expressed genes (DEGs) from 3D_MyoD, 3D+FI, and 3D_MyoD+FI compared to the original 3D_fibroblasts. (**E**) Heat map showing the representative genes differentially expressed between 3D_MyoD+FI and

*Figure 8 continued on next page*

*Figure 8 continued*

3D_fibroblast cells (n = 3 biologically independent samples). (**F**) Gene Ontology (GO) analysis of upregulated DEGs between 3D_MyoD+FI vs. 3D_fibroblast cells. (**G**) GOChord analysis of the upregulated genes within representative pathway between 3D_MyoD+FI and 3D_fibroblast cells.

were simultaneously activated during myogenic/adipogenic transdifferentiation, which should have profound effects on both myogenesis (*Bakkar and Guttridge, 2010*; *Jang and Baik, 2013*; *Keren et al., 2006*) and adipogenesis (*Batista et al., 2012*; *Bost et al., 2005*; *Richard and Stephens, 2011*; *Figure 8F*). The upregulated genes in the representative pathways (such as '*Stat1*' in JAK-STAT signaling pathway, '*Mapkapk3*' in MAPK signaling pathway) are shown in *Figure 8G*. In conclusion, the transcriptome analysis of the different types of transdifferentiated cells revealed important molecular mechanisms including not only the myogenic- and adipogenic-specific pathways driving the muscle formation and fat deposition respectively, but also several key multifunctional signaling pathways that can promote the cell fate transition and differentiation in different cellular contexts including the muscle and fat tissues.

## Discussion

Mature muscle tissue primarily consists of myofibers (myotubes), which are long, multinucleated cells that contract to generate force and movement. In addition to myofibers, muscle tissue contains a variety of other cell types, including fibroblasts and adipocytes, all of which play important roles in the structure and function of the tissue. The ECM, which is mainly secreted by fibroblast cells, provides support and structural integrity to the muscle tissue. It is made up of a complex network of proteins and molecules, such as collagen, that provide a scaffold for the cells to attach to and interact with (*Franco-Barraza et al., 2016*). Together, these components contribute to the unique features of the skeletal muscle tissue and the fresh meat, including its strength, flexibility, and elasticity. It has been and still is challenging to recreate those characteristics in an in vitro condition of cultured meat production. One of the main obstacles in this process is the co-culturing of different cell types with distinct properties. In this study, we overcame this limitation of co-culturing different types of cells by utilizing a single-source cell to generate various meat components, including muscle, fat, and collagen. Precisely, we employed chicken fibroblasts to produce muscle, deposit fat, and synthesize collagen in a well-controlled and adjustable manner within a 3D setting to produce meat with desirable characteristics.

Fibroblasts are one of the most common cell types in animals and could serve as the seed cells for cultured meat production due to their unique and versatile features. First, the fibroblast cells are widely available in the bodies of agricultural animals and could be easily isolated and cultured in vitro. Second, many groups have successfully transformed the chicken fibroblasts into immortalized cell lines (*Himly et al., 1998*; *Pasitka et al., 2023*). These cell lines can provide an unlimited cellular resource for cultured meat production and eliminate the need for animal or embryo harvest. Third, fibroblasts are capable of adapting to low serum concentration medium or even serum-free medium, which greatly reduce the culturing cost and risk of serum-bound pathogens (*Genbacev et al., 2005*; *Lohr et al., 2009*; *Pasitka et al., 2023*). Lastly, the fibroblast cells can also undergo transformation to enable high-density propagation in suspension culture (*Bürgin et al., 2020*; *Fluri et al., 2012*; *Shittu et al., 2016*), which is a crucial step toward scaling up to mass production. The Food and Drug Administration (FDA) has already approved the use of chicken fibroblast cells for cultured meat production, and recently, Eat Just Inc successfully utilized chicken fibroblasts for the commercial production and sale of cultured meat in Singapore and the USA (FDA, 2023). One latest study has also demonstrated the feasibility and consumer acceptance of cultured meat derived solely from native chicken fibroblast cells, indicating a very promising development. However, the resulting meat products do not seem to contain any muscle components (*Pasitka et al., 2023*). We have previously established an effective strategy for myogenic transdifferentiation, allowing for the production of muscle cells from fibroblast cells of various species including chicken and pig in 2D culture (*Ren et al., 2022*). In the present study, we further enhanced the myogenic transdifferentiation process in 3D and simultaneously simulated the fat deposition to create cell-based meat that more closely resembles real meat. As a proof of concept, we utilized the transgene method to achieve maximum myogenic induction and the final products still retain the foreign transgene fragment in the cells' genome. It is

therefore posing a risk of genetic modified food that is not suitable for mass production. In the next step, other non-transgenic means such as non-integrating vectors, chemical reprogramming, modified RNAs, and recombinant transgene removal techniques will be explored to develop transgene-free end products. Another food safety concern in this study is the use of GelMA hydrogel for culture meat production. Due to its excellent biocompatibility and mechanical flexibility, GelMA-based hydrogel has demonstrated significant potential in scalable 3D cell culture for creating artificial tissue ranging in sizes from millimeters to centimeters. It is widely used in 3D cell culture and tissue engineering for regenerative medicine, but less common in food processing and agricultural applications. Due to its special photo-crosslinking properties, biocompatibility, and degradability, it allows this material to be shaped into complex tissue structures by 3D printing or modeling. Many researchers have also used GelMA hydrogel as a scaffold for culture meat production (*Jeong et al., 2022*; *Zheng et al., 2021*). Later research will carefully consider hydrogel as well as other types of scaffold biomaterials for cost-effective and food-safety-compliant culture meat production (*Bomkamp et al., 2022*).

Numerous studies have identified the crucial role of fat in the aroma, juiciness, and tenderness of meat. In general, a very low level of intramuscular fat results in dry meat with a plain taste, whereas the high intramuscular fat contents can improve the cooking flavor and greatly increase the value of meat products, such as the high marbling beef from Japanese Wagyu cattle (*Gotoh et al., 2018*; *Motoyama et al., 2016*). The fat content of fresh meat mainly comes from lipids contained in the fat cells. Thus, closely mimicking the intramuscular fat properties in cultured meat would require co-culturing of muscle cells with fat cells. For example, co-culturing pre-adipocytes with myoblasts may increase the intramuscular fat content, tenderness, and taste intensity of cultured meat (*Lau et al., 1996*; *Pandurangan and Kim, 2015*; *Zagury et al., 2022*). However, co-culture of different cell types is technically challenging since each cell type grows and differentiates in the specific optimized medium. When different cell types are cultured in the same medium, these culture conditions may be suboptimal for one cell type or the other and result in inefficient cellular growth (*Pallaoro et al., 2023*). Previous studies have underlined the influence that adipocytes growing near muscle cells can impair myogenesis (*Seo et al., 2019*; *Takegahara et al., 2014*). Simultaneous or sequential induction of both myogenic and lipogenic differentiation in the same starting seed cells would resolve these co-culture conflicts, and the multi-lineage competent chicken fibroblast cells were chosen to explore this double transdifferentiation strategy. As a proof of concept, we successfully transformed chicken fibroblast cells into muscle cells and deposited fat into the same cells in 3D hydrogel matrix. Notably, the intramuscular fat content in the cultured meat could be tailored to any specific level within a certain range. From a nutritional point of view, a direct comparison with the traditional chicken meat was performed. The triglyceride content in the cultured meat is comparable to that of chicken meat,

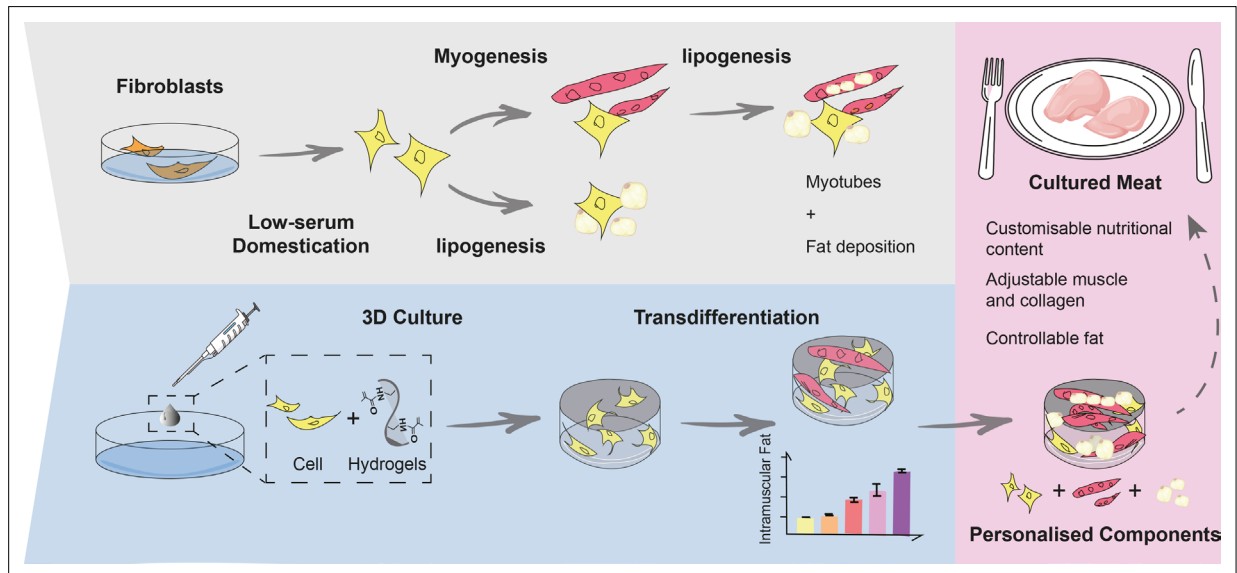

**Figure 9.** Model for myogenic and lipogenic transdifferentiation of chicken fibroblasts in 3D culture to produce meat with precisely controlled levels of intramuscular fat and extracellular matrix.

and, more importantly, the amount of fat could be easily manipulated in order to achieve a more attractive nutritional profile (*Figure 9*).

In this study, the deposition of fat in the myotubes/myofibers facilitated the storage of significant lipid quantities in transdifferentiated muscle cells, known as intramyocellular lipids. Additionally, we observed Oil Red O staining in the remaining un-transdifferentiated fibroblasts, resembling cells of intramuscular adipocytes (extramyocellular lipids) found within muscle tissue. Hence, current adipogenic induction treatment caused lipogenesis in both the MyoD-transdifferentiated cells and un-transdifferentiated fibroblasts. At present, we do not know whether the fatty acids profile would be different between the two types of origins. Unsaturated fats are conventionally regarded as 'healthier' than saturated fats (*Berglund et al., 2007*; *Hamley, 2017*). One of the superior assets of cultured meat production is the ability to increase the unsaturated fat levels by adjusting the additive fatty acids, oleic/linoleic acids in this case, in the culture medium. In the future, further comprehensive analysis of the fatty acids profile in the cultured meat is required to elucidate the fat-associated attributes similar or distinct to real meat.

The current transcriptome analysis during the cellular transdifferentiation also revealed that key regulatory signaling pathways control the formation of cultured meat derived from fibroblasts. Notably, we found that several multifunctional pathways such as JAK-STAT and MAPK signaling pathways were significantly enriched in the final cells that underwent double myogenic/adipogenic transdifferentiation. These signalings could drive the differentiation process of many different types of cells, including muscle and adipose tissues (*Bost et al., 2005*; *Jang and Baik, 2013*; *Keren et al., 2006*; *Richard and Stephens, 2011*), and may be manipulated in more settings to further improve the generation of final cultured meat with tunable fat content.

In conclusion, we have effectively utilized immortalized chicken fibroblasts in conjunction with classical myogenic/adipogenic transdifferentiation approaches within the 3D hydrogel to establish a cultured meat model. This model allows for the precise regulation of the synthesis of key components of conventional meat, including muscle, fat, and ECM. This approach can be readily extrapolated to other species such as pigs and cows, and presents promising avenues for the large-scale production of customized and versatile meat products that may cater to varying consumer preferences.

## Materials and methods

### Cell preparation and inducible myogenic transdifferentiation

The cellular transdifferentiation was constructed as described previously (*Ren et al., 2022*; *Ren et al., 2023*). Briefly, we cloned the chicken full MyoD coding sequence fused in-frame with 3xFlag into a DOX-inducible lentiviral system (Tet-On-MyoD). The wild-type chicken fibroblasts were infected with lentivirus and subjected to puromycin selection, and finally obtained the transdifferentiation cell lines. In addition, myofibroblasts were isolated from the skin of 10-day-old chicken embryos in the same way as in the previous study (*Kosla et al., 2013*). Cells were cultured in 1640 basal medium (Gibco, #C11875500BT) supplemented with 12% FBS (CELLiGENT, #CG0430A) and 1% penicillin-streptomycin (Gibco, #11140050) at 39°C under 5% $CO_2$ atmosphere and were given fresh medium every 2 d. When grown to approximately 80% confluence, the cells were trypsinized and passaged. The study was approved by the Animal Care and Use Committee of Shandong Agricultural University.

### Domestication of cells in low-concentration serum medium

The chicken fibroblast cells were domesticated with the progressive reduced concentrations of serum. In general, cells were cultured with 12% FBS in 1640 basal medium, and when grown to about 80% confluence, the medium was replaced with a 6% FBS medium for further cell culture and passage. The medium was then changed to 3% FBS medium for further cell culture and passage depending on the cell status, and so on. This was started again and the above steps were repeated as soon as the cells grew badly or died. We directly changed the cell culture medium from 12% FBS to 2% CS then reduced it to 1% CS when using CS (Solarbio, #S9080) medium and the cells could adapt to the medium with low concentration of CS after 3–5 passages.

## Preparation of 3D scaffold and cell culture in 3D matrix

GelMA hydrogels were purchased from Beijing ShangPu for this experiment. Appropriate amount of GelMA powder was weighed, dissolved in 1640 basal medium on a water bath at 70°C for about 30 min, and then filtered with 0.22 µm sieve. Then, 1/8 volume of lithium acylphosphinate salt photoinitiator was added to the dissolution solution to obtain GelMA hydrogel solution, then stored at 37°C until usage but no more than 24 hr. Fibroblast cells were obtained by trypsin treatment and suspended with GelMA hydrogel solution and gently mixed, followed by treatment of 405 nm UV light for 10–20 s to get a 3D hydrogel scaffold. In addition, the hydrogel was secured on a fixation ring for better cell growth and easy movement (*Figure 3—figure supplement 3*). The cell hydrogel complexes were placed in 24-well plate and culture with 1640 medium containing 12% FBS. Then, they were transferred to a new well after 12 hr, and fresh medium was added and changed every 24 hr. The cell hydrogel complexes were gently washed three times with PBS buffer and digested by collagenase II enzyme (Worthington, LS004177) in an incubator at 39°C for 6 min until the cells in the hydrogel package became round and shed. The digest was terminated by adding medium and then centrifuged at 1000 × *g* for 10 min at room temperature (25 °C).

## Cell Counting Kit-8 assay

Cells cultured in 1% CS and 12% FBS were seeded into 96-well plates with 100 µl of medium per well and incubated for 0 hr, 24 hr, 48 hr, and 72 hr. 10 µl of the Cell Counting Kit-8 assay (CCK-8) solution (Solarbio, #CA1210, China) was then added to each well and incubated for 2 hr. The absorbance values at 450 nm were measured with an EnSpire multifunctional spectrophotometer (PerkinElmer, USA).

## EdU assay

2D cells were cultured in 6-well plates, and 1 ml of the growth medium was added to each well with 0.25 µl of 1.25 mg/ml 5-ethynyl-2'-deoxyuridine (EdU) (Beyotime, #ST067-1g). After 30 min of incubation, cells were fixed by 4% paraformaldehyde (PFA) for 30 min. The cells were stained with a prepared reaction solution consisting of 1 mmol/L $CuSO_4$, 100 mmol/L Tris–HCl, 100 mmol/L ascorbic acid, 1:1000 Alexa Fluor 555 Azide as previous described (*Zhang et al., 2022*). After 30 min, the cells were washed with PBS, and the nuclei were stained with 50 ng/ml DAPI for 10 min. 3D cultures of cells were cultured in 24-well plate by adding 1 ml of the medium with 0.25 µl EdU, incubated for 1 hr, and then fixed with 4% PFA for 24 hr. Staining time was extended to 1 hr and nuclear staining to 20 min. Fluorescence images were collected using fluorescence microscopy.

## Cell differentiation

For fibroblast induction into myoblasts, 50 ng/ml DOX (Sigma, #D3000000) in 1640 basal medium containing 12% FBS was added for 3 d, and the differentiation medium was replaced when the cells reached 80% confluence. In addition, in 3D culture, cells were proliferated for 7 d before changing the differentiation medium when a dense arrangement of cells can be observed under the microscope.

For induction of lipogenic differentiation, the differentiation medium was changed when the fusion rate of cells reached 80% in 2D culture or proliferated for 5–7 d in 3D culture. The lipogenesis was induced for 48 hr with fresh medium changes every 24 hr. For the differentiation experiments, 1640 with 12% FBS was used as a control, and the lipogenic medium was consistent with a 1:100 fatty acid ('F' for short) composition of 1:1 oleic acid (Sigma, #O3008, 2 mol oleic acid/mole albumin) and linoleic acids (Sigma, #L9530, 2 mol linoleic acid/mole albumin; 100 mg/ml albumin), and insulin ('I' for short) (Sigma, #I0516) concentration of 60 µg/ml.

## Immunofluorescence staining

Similar to our previous steps for immunofluorescence staining of cells in 2D culture (*Luo et al., 2022*), cells were fixed in 4% PFA for 30 min and washed three times with PBS buffer at room temperature, then permeabilized with 0.5% Triton X-100 in PBS for 10 min and blocked with 10% goat serum at 0.5% TritonX-100 for 1 hr. Primary antibodies were diluted with 10% goat serum in 0.5% Triton X-100 at 4°C for 12 hr. The primary antibodies for MHC (DSHB, #AB2147781), Desmin (Sigma, #D8281) and laminin (Sigma, #F1804) were added at a dilution of 1:500, and the primary antibodies for Vimentin (DSHB, #AB528504) was added at a dilution of 1:200. This was followed by incubation with Alexa

secondary antibody (Invitrogen, #A-21202, #A-21206, and #A-31570) at a 1:500 dilution for 2 hr at room temperature. The antibody for α-SMA (Sigma, C6198) was added at a dilution of 1:500 and incubated with 10% goat serum in 0.5% Triton X-100 at room temperature for 2 hr. Nuclear staining was performed with 50 ng/ml DAPI (Sigma, #D8417) for 10 min. Cells in the 3D culture were fixed for 48 hr, the total time of permeabilizing and the blocking was no more than 24 hr, and the incubation time of the primary antibody was extended to 24 hr, the secondary antibody to 4 hr, and the nuclear staining to 20 min. For MHC detection, fluorescence images were collected using fluorescence microscopy, and the MHC$^+$DAPI$^+$/DAPI$^+$ differentiation index was calculated from three or more images using confocal microscopy (Zeiss LSM 800).

### Oil Red O staining

For Oil Red O staining, cells were fixed with 4% PFA for 30 min or 24 hr, respectively, in 2D and 3D culture. After washing thrice with PBS at room temperature, the cells were soaked in 60% isopropanol for better coloration of Oil Red and then washed for 5 min or 10 min, respectively, in 2D and 3D culture. The cells were stained with Oil Red O (Sigma, #O0625) for 30 min or 60 min, respectively, in 2D and 3D. The cells were then washed with 60% isopropanol for 30 s or 1 min, respectively, in 2D and 3D to remove surface staining. Then, the cells were washed with distilled water three times and the stained lipid droplets were visualized using a microscope.

### mRNA extraction and RT-qPCR

Cells were lysed in Trizol (Simgen, #5301100) and RNA was extracted following the manufacturer's recommendations. RNA concentration was measured on NanoDrop2000 (Thermo Scientific, USA). 1 µg of RNA was reverse-transcribed using PrimeScript RT reagent kit (Takara, #RR047A). Real-time quantitative PCR was performed using SYBR Green Mix (Abclonal, #RK21203) following the manufacturer's instructions. Expression was normalized to GAPDH using delta-delta-CT method. For

**Table 1.** List of primers of qPCR.

| Gene | Forward primer | Reverse primer |
|---|---|---|
| Gapdh | TCGGAGTCAACGGATTTGGC | ATAGTGATGGCGTGCCCATT |
| MyoD | ACTACAGCGGGGAGTCAGAT | GCTTCAGCTGGAGGCAGTAT |
| MyoG | AGCCTTCGAGGCTCTGAAAC | AAACTCCAGCTGGGTGCTC |
| Myh15 | AGATAAAGGAACTACAGGCTCGT | CGCCAGCTTCAGGAACTCA |
| CKM | ACCTGGACCCCAAATACGTG | TCGAACAGGAAGTGGTCGTC |
| Desmin | GGAGATCGCCTTCCTCAAGA | CAGGTCGGACACCTTGGATT |
| Six1 | ACTGCTTCAAGGAGAAGTCG | TTCTCCGTGTTCTCCCTCTC |
| Thy-1 | TGTCATCCTGACAGTGCTGC | GGTAGAGGCACACCAGGTTC |
| TGFβ–1 | GAGCTGTACCAGGGTTACG | GAAGCCTTCGATGGAGATG |
| TGFβ–3 | CTCCCCGAGCACAATGAGT | TATATGCTCATCTGGCCGCA |
| Smad3 | GCAAGATCCCACCAGGATG | GAGGTGCAGCTCAATCCAG |
| Pparg | TGCCAAGCATTTGTAT | TGCGAATTGCTACTTCTTTGTT |
| Znf423 | CCAGTGCCCACAGAAGTTCT | CCACTGTGCCACCATCAAGT |
| Fabp4 | CAAGCTGGGTGAAGAGTTTGATG | TCGTAAACTCTTTTGCTGGTAAC |
| Gpd1 | GGCTTTTGCCAAGACTGGGAA | GGTTTGCCCTCATAGCAGATCTG |
| Collagen I α1 | GTCCTGCTGGATTTGCTGG | GAAACCAGTAGCACCAGGG |
| Collagen I α2 | TGATCCATCTAAAGCGGCTG | TTTGCCAGGGTGACCATCTT |
| Laminin | CGCGATTTCTGATTTTGCCG | CATTGCAGTCACAAGGCAAG |
| Fibronectin | GTGCTACGACGATGGGAAAA | GCAGTTGACGTTGGTGTTTG |
| Elastin | CTACTGGGACAGGTGTTGGA | CACCATAGGCTCCTGCCTT |

comparisons of the expression, we used a one-tailed Student's *t*-test. The error bars indicate the SEM. The RT-qPCR primers are described in *Table 1*.

## Transcriptome analysis

The RNA-seq library-preparation protocol was based on the NEBNext Ultra RNA Library Prep Kit for Illumina (NEB, #E7530L). Insert size was assessed using the Agilent Bioanalyzer 2100 system and qualified insert size was accurate quantification using StepOnePlus Real-Time PCR System (Library valid concentration＞10 nM) and then paired-end sequencing using an Illumina platform. RNA-seq was performed on triplicates for each sample. Sequencing data have been deposited in SRA database under accession code PRJNA1102033.

## RNA-seq data analysis

The RNA-seq raw data were first trimmed adapters by trim_galore software (*Bolger et al., 2014*) and then the clean data of RNA-seq was mapped to the chicken genome (Ensembl, GRCg7b) using hisat2 (*Kim et al., 2019*) with default parameters. Because of using paired-end reads, the concordant unique mapping reads/pairs were kept based on the mapping flags. The duplications were removed based on the coordinates of the reads/pairs. The de-duplication unique mapping reads/pairs were used for further analysis in this study. The read counts for each sample were computed with the featureCounts (*Liao et al., 2014*) software, and the RefSeq gene annotation for chicken genome assembly is GRCg7b. The transcript per million (TPM) was normalized using the read counts. DEGs of RNA-seq data were analyzed using DESeq2 called by q<0.01 and fold change >2 thresholds. The chicken genes were transformed to homolog human genes using the ensemble bioMart database, the GO in this study was conducted in Metascape, and the enriched top pathways are shown. All plots were generated with R (v4.0.3).

## Measurement of total collagen content in cultured meat

Total collagen content was determined by the concentration of hydroxyproline. The medium was replaced with fresh medium in the incubator for 2 hr before the assay and washed three times with PBS. According to the instructions of the Hydroxyproline Assay Kit (Jiancheng, #A030-1-1, China), 1 volume of saline was added to dried cultured meat at a 1:1 ratio of weight and volume. After mechanical homogenization in an ice-water bath, the digestion solution was incubated at 37°C in a water bath for 4 hr. Hydroxyproline content was measured by the absorbance at 550 nm using an EnSpire multifunctional spectrophotometer (PerkinElmer). For each measurement of cultured meat, the hydroxyproline content of the corresponding blank cell-free hydrogel was subtracted. The amount of collagen was calculated from the hydroxyproline concentration with a conversion factor of 7.25 in μg/mg wet tissue (*Vasanthi et al., 2007*; *Zheng et al., 2021*).

## Measurement of triglyceride content in cultured meat

The triglyceride content in cultured meat was measured using the kit (Solarbio, #BC0620, China). Before the assay, the culture system was washed three times with PBS and uniformly added to the same medium containing 12% FBS in an incubator for 2 hr. It was then removed and washed three more times with PBS. The cultured meat was taken out and churned, and the precipitate is obtained by centrifugation, which should be well air-dried. The precipitate was weighed and Trizol was added to lyse cell for 2 hr. A mixture of n-heptane and isopropanol at a ratio of 1:1 was then added, shaken, and mixed. Then, potassium hydroxide was added and fullly shaken to produce glycerol and fatty acids, and the other reagents were added in sequence according to the instructions. The final triglyceride content was measured by the specific light intensity at 420 nm, as described previously.

## Emission scanning electron microscope

The cell hydrogels were washed three times with PBS and then fixed in 4% PFA for 1 hr. The samples were randomly clamped into spiking trays, rapidly frozen in liquid nitrogen, and then the images were randomly collected using the emission scanning electron microscope (Hitachi SU8010, Japan).

## Statistical analyses

Statistical analyses were performed using the GraphPad Prism software. For normally distributed data sets with equal variances, a two-sample $t$-test was used. The significance of differences is provided in the figure legends.

## Acknowledgements

This work was supported by the National Natural Science Foundation of China (31771617), Postdoctoral Fellowship Program of CPSF (GZC20230775), Taishan Scholar Program (202211100), Scientific Research Innovation Team of Young Scholar of Shandong, and HZAU-AGIS Cooperation Fund (SZYJY2021009).

## Additional information

### Funding

| Funder | Grant reference number | Author |
| --- | --- | --- |
| National Natural Science Foundation of China | 31771617 | Heng Wang |
| China Postdoctoral Science Foundation | GZC20230775 | Ruimin Ren |
| Taishan Scholar Program | 202211100 | Heng Wang |
| Scientific Research Innovation Team of Young Scholar of Shandong | | Heng Wang |
| Huazhong Agricultural University | SZYJY2021009 | Heng Wang |
| Chinese Academy of Agricultural Sciences | SZYJY2021009 | Heng Wang |

The funders had no role in study design, data collection and interpretation, or the decision to submit the work for publication.

### Author contributions

Tongtong Ma, Conceptualization, Data curation, Formal analysis, Investigation, Methodology, Writing – original draft; Ruimin Ren, Conceptualization, Data curation, Investigation, Methodology; Jianqi Lv, Data curation, Formal analysis, Investigation, Methodology; Ruipeng Yang, Xinyi Zheng, Guiyu Zhu, Investigation; Yang Hu, Resources, Methodology; Heng Wang, Conceptualization, Supervision, Funding acquisition, Writing – original draft, Project administration, Writing - review and editing

### Author ORCIDs

Heng Wang ⓘ http://orcid.org/0000-0002-1727-9226

Reviewer #1 (Public Review): https://doi.org/10.7554/eLife.93220.3.sa1
Reviewer #2 (Public Review): https://doi.org/10.7554/eLife.93220.3.sa2
Author response https://doi.org/10.7554/eLife.93220.3.sa3

## Additional files

### Supplementary files
• MDAR checklist

### Data availability
Sequencing data have been deposited in NCBI BioProject under accession code PRJNA1102033.

The following dataset was generated:

| Author(s) | Year | Dataset title | Dataset URL | Database and Identifier |
|---|---|---|---|---|
| Ren R, Wang H | 2024 | Transdifferentiation of fibroblasts into muscle cells to constitute cultured meat with tunable intramuscular fat deposition | https://www.ncbi.nlm.nih.gov/bioproject/PRJNA1102033/ | NCBI BioProject, PRJNA1102033 |

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
