## [Editor Report · eLife assessment]

This study presents an **important** new technology for transdifferentiation of fibroblasts into muscle cells. The data and methods used for analysis were **compelling**. This study will have broad interest to cellular reprogramming biologists in particular as well as the general public.

---

## [Referee Report · Reviewer #1 (Public Review)]

Summary:

The authors presented here a novel 3D fibroblast culture and transdifferentiation approach for potential meat production with GelMA hydrogel.

Strengths:

(1) Reduced serum concentration for 3D chicken fibroblast culture and transdifferentiation is optimized.

(2) Efficient myogenic transdifferentiation and lipogenesis as well as controlled fat deposition are achieved in the 3D GelMA.

---

## [Referee Report · Reviewer #2 (Public Review)]

The manuscript by Ma et al. tries to develop a protocol for cell-based meat production using chicken fibroblasts as three-dimensional (3D) muscle tissues with fat accumulation. The authors used genetically modified fibroblasts, which can be forced to differentiate into muscle cells, and formulated 3D tissues with these cells and a biphasic material (hydrogel). The degrees of muscle differentiation and lipid deposition in culture were determined by immunohistochemical, biochemical, and molecular biological evaluations. Notably, the protocol successfully achieved the process of myogenic and lipogenic stimulation in the 3D tissues.

As addressed after the initial review process, the manuscript is clearly written with well-supportive figures. The study design is reasonable with adequate analysis. In the revised manuscript, the authors further discussed the ideas in terms of the approach using genetic modification for cell-based meat production. However, more careful considerations may still be helpful when actually using the technology for cultivated meat production.

---

## [Author Response]

The following is the authors’ response to the original reviews.

**eLife assessment**
This solid study investigates the transdifferentiation of chicken embryonic fibroblasts into muscle and fat cells in 3D to create whole-cut meat mimics. The study is important and provides a method to control muscle, fat, and collagen content within the 3D meat mimics and thus provides a new avenue for customized cultured meat production. Limitations of this study include the use of transgene for transdifferentiation and thus the creation of GMO food.

We are grateful for the substantial effort that editors and reviewers put into assessing our manuscript and providing insightful feedback. We have tried to address, as much as possible, all comments and criticisms. We believe that we have now a significantly improved manuscript. Below, there is a point-by-point response.

**Public Reviews:**

**Reviewer #1 (Public Review):**
Summary:The authors presented here a novel 3D fibroblast culture and transdifferentiation approach for potential meat production with GelMA hydrogel.Strengths:(1) Reduced serum concentration for 3D chicken fibroblast culture and transdifferentiation is optimized.(2) Efficient myogenic transdifferentiation and lipogenesis as well as controlled fat deposition are achieved in the 3D GelMA.Weaknesses:(1) While the authors stated the rationale of using fibroblasts instead of myogenic/adipogenic stem cells for meat production, the authors did not comment on the drawbacks/disadvantages of genetic engineering (e.g., forced expression of MyoD) in meat production.

Thanks for the reviewer for raise this important issue. We have now described this drawback in the discussion part.

As a proof-of-concept study, we sought to explore the potential of utilizing the transdifferentiation integrated transgene tools for overexpressing a transdifferentiation factor to achieve the maximum muscle production. However, it is important to acknowledge that genetically modified meat products derived from the genetic engineering of cultured cells will not be suitable for consumer acceptance and market viability. We are currently testing other non-genomic integrating delivery means such as modRNAs and chemical cocktails to induce myogenic transdifferentiation in fibroblasts. We believe the new non-genomic integration means would be compatible for the meat production and consumer acceptance.

Please see lines 439-445.

“As a proof-of-concept, we utilized the transgene method to achieve maximum myogenic induction and the final products still retain the foreign transgene fragment in the cells’ genome. It is therefore posing a risk of genetic modified food which is not suitable for mass production. In the next step, other non-transgenic means such as non-integrating vectors, chemical reprogramming, modified RNAs, and recombinant transgene removal techniques will be explored to develop transgene-free end products.”

(2) While the authors cited one paper to state the properties and applications of GelMA hydrogel in tissue engineering and food processing, concerns/examples of the food safety with GelMA hydrogel are not discussed thoroughly.

Thank you for pointing out this issue. We discussed the drawbacks of Gelma hydrogel applications in the meat production in the main text.

GelMA-based hydrogels have shown great potential due to their biocompatibility and mechanical tenability. It is widely used in 3D cell culture and tissue engineering for regenerative medicine, but less common in food processing and agricultural applications. Due to its special photo-crosslinking properties, biocompatibility and degradability, it allows this material to be shaped into complex tissue structures by 3D printing or modelling. Many researchers have also used Gelma hydrogel as a scaffold for culture meat production (Jeong et al., 2022; Li et al., 2021; Park et al., 2023). Later research will carefully consider Gelma hydrogen as well as other types of scaffold biomaterials for cost-effective and food-safety compliant culture meat production (Bomkamp et al., 2022).

Bomkamp, C., Skaalure, S. C., Fernando, G. F., Ben‐Arye, T., Swartz, E. W., & Specht, E. A. J. A. S. (2022). Scaffolding biomaterials for 3D cultivated meat: prospects and challenges. Advanced Science (Weinh), 9(3), 2102908.

Jeong, D., Seo, J. W., Lee, H. G., Jung, W. K., Park, Y. H., & Bae, H. (2022). Efficient Myogenic/Adipogenic Transdifferentiation of Bovine Fibroblasts in a 3D Bioprinting System for Steak-Type Cultured Meat Production. Advanced Science (Weinh), 9(31), e2202877.

Li, Y., Liu, W., Li, S., Zhang, M., Yang, F., & Wang, S. J. J. o. F. F. (2021). Porcine skeletal muscle tissue fabrication for cultured meat production using three-dimensional bioprinting technology. Journal of Future Foods, 1(1), 88-97.

Park, S., Hong, Y., Park, S., Kim, W., Gwon, Y., Jang, K.-J., & Kim, J. J. J. o. B. E. (2023). Designing Highly Aligned Cultured Meat with Nanopatterns-Assisted Bio-Printed Fat Scaffolds. Journal of Biosystems Engineering, 48(4), 503-511.

We discussed the drawbacks of GelMA hydrogel. Please see lines 445-457.

“Another food safety concern in this study is the use of GelMA hydrogel for culture meat production. Due to its excellent biocompatibility and mechanical flexibility, GelMA-based hydrogel has demonstrated significant potential in scalable 3D cell culture for creating artificial tissue ranging in sizes from millimeters to centimeters. It is widely used in 3D cell culture and tissue engineering for regenerative medicine, but less common in food processing and agricultural applications. Due to its special photo-crosslinking properties, biocompatibility and degradability, it allows this material to be shaped into complex tissue structures by 3D printing or modelling. Many researchers have also used GelMA hydrogel as a scaffold for culture meat production (Jeong et al., 2022; Li et al., 2021; Park et al., 2023). Later research will carefully consider hydrogel as well as other types of scaffold biomaterials for cost-effective and food-safety compliant culture meat production (Bomkamp et al., 2022). ”

(3) In Fig. 4C, there seems no significant difference in the Vimentin expression between Fibroblast_MyoD and Myofibroblast. The conclusion of "greatly reduced in the myogenic transdifferentiated cells" is overstated.

Thanks for pointing out this mistake.

We revised the wording accordingly. The vimentin expression was reduced in fibroblast_MyoD compare to the original fibroblast.

Please see lines 231-233.

“The fibroblast intermediate filament Vimentin (Tarbit et al., 2019) was abundantly expressed in the fibroblasts but reduced in the myogenic transdifferentiated cells (Figure 4C)”

(4) The presented cell culture platform is only applied to chicken fibroblasts and should be tested in other species such as pigs and fish.

Thank you for the suggestion.

In this pilot cultured meat study, we utilized chicken embryonic fibroblasts. These specific cells were chosen for their near-immortal nature and robustness in culture, as well as the inducible myogenic capacity. In our previous experiments (Ren et al, Cell Reports, 2022, 40:111206), we have tested the myogenic transdifferentiation potential of fibroblasts from mice, pigs, and chickens, and observed varying efficiencies of myogenesis. It is important to note that fibroblast cells derived from different species, or even different tissues within the same species, would exhibit significant variations in their capacities for myogenic and adipogenic transdifferentiation.

In this proof-of-concept study we used only one source of fibroblasts for testing culture meat production and confirmed the myogenic/adipogenic transdifferentiation could be manipulated as feasible means to precisely control muscle, fat and collagen content. We would expect that different origins of fibroblasts to display different transdifferentiation efficiencies and thus produce various muscle/fat ratios in meat mimics. That is beyond the scope of current study.

Furthermore, we are also testing myogenic/adipogenic transdifferentiation of fibroblasts from pigs through non-genomic integration approaches. We believe only the non-transgene tools are viable solutions for culture meat production in the future. We added the species information in the discussion part.

See lines 515-517.

“This approach can be readily extrapolated to other species such as pigs and presents promising avenues for the large-scale production of customized and versatile meat products that may cater to varying consumer preferences.”

**Reviewer #2 (Public Review):**
The manuscript by Ma et al. tries to develop a protocol for cell-based meat production using chicken fibroblasts as three-dimensional (3D) muscle tissues with fat accumulation. The authors used genetically modified fibroblasts which can be forced to differentiate into muscle cells and formulated 3D tissues with these cells and a biphasic material (hydrogel). The degrees of muscle differentiation and lipid deposition in culture were determined by immunohistochemical, biochemical, and molecular biological evaluations. Notably, the protocol successfully achieved the process of myogenic and lipogenic stimulation in the 3D tissues.Overall, the study is reasonably designed and performed including adequate analysis. The manuscript is clearly written with well-supported figures. While it presents valuable results in the field of cultivated meat science and skeletal muscle biology, some critical concerns were identified. First, it is unclear whether some technical approaches were really the best choice for cell-based meat production. Next, more careful evaluations and justifications would be required to properly explain biological events in the results. These points include additional evaluations and considerations with regard to myocyte alignment and lipid accumulation in the differentiated 3D tissues. The present data are very suggestive in general, but further clarifications and arguments would properly support the findings and conclusions.

Thanks for the reviewer’s comments. We have performed additional experiments and analysis to address the critical questions. We also revised the text extensively to clarify or discuss some of the concerns, such as the cell alignment and cellular distribution of intramuscular fat issues. We expect the revised data and text could adequately support the conclusions of the manuscript.

**Recommendations for the authors:**

**Reviewer #1 (Recommendations For The Authors):**
(1) In Figure 1, the authors used 1% chicken serum. Have the authors tested other lower concentrations? It will be interesting to see the lowest chicken serum concentrations in fibroblast culture and transdifferentiation;

Thank you for your suggestion.

Yes, we actually have tested the lower concentrations of serum, such as 1% FBS, and 0.5% chicken serum. However, the cells are not in a healthy state under these low levels of serum, as shown by the abnormal cell morphology and nearly no cell growth. Please see the revised Supplementary Figure S1D, in which we added the 1%FBS and 0.5% chicken serum data. Hence, the 1% chicken serum is optimal in our hands. We will also test other types of specialized serum-free medium in future experiments.

(2) In Figure 2, the authors should quantify the fold expansion of fibroblasts cultured in 3D gel after 1, 3, 5, and 9 days since this data is important for future meat manufacturing. In addition, long-term expansion (e.g., 1 month) in 3D gel should also be shown;

Thanks for the question. We have quantified the cell growth in 3D by measuring the PHK26 stained cells. Since the cells were implanted into the gel, they propagated exponentially from 1 day to 9 days. The cell proliferation data provide good reference for the future meat manufacturing (Figure 2D). We have tried the long-term expansion in 3D but failed to measure the cell proliferation. Because the 3D gel always collapsed during 12-15 days in cell culture for some unknown reasons, either the cells are grown too crowded to compromise the gel structure or the gel matrix itself is not strong enough for standing long-term. We believe the cells will grow well in long-term if we provide enough 3D attachment surface, since they grow indefinitely in 2D. We will testing different 3D matrix in the future.

Please see the revised Figure 2D for the quantification of cells.

(3) In Figure 3, please also show MyoD staining as it'll be interesting to see the expression of exogenous and endogenous MyoD expression after dox treatment. In Figure G, the hydrogel meat seems very small, please show/discuss the maximum size of hydrogel meat that may be achieved using this approach;

Thanks for asking this information. We performed the immunostaining by using the anti-MyoD and anti-Flag to show the expression of all MyoD (exogenous and endogenous) and only exogenous MyoD after dox treatment. The MyoD and 3xFlag were fused in-frame in the transgene plasmid and thus the anti-Flag staining indicate the exogenous MyoD expression and anti-MyoD staining indicate the expression of exogenous and endogenous MyoD together.

As shown in Figure S4, we found that almost 100% of cells were positive for MyoD staining and 60% of which expressed Flag, these data were consistent with our previous results (Ren et al., 2022, Cell Reports).

As for the size of the culture meat based on hydrogel, we discussed the possibilities in scalable production of hydrogel based whole-cut meat mimics.Please see lines 446-449.“Due to its excellent biocompatibility and mechanical flexibility, GelMA-based hydrogel has demonstrated significant potential in scalable 3D cell culture for creating artificial tissue ranging in sizes from millimeters to centimeters.”

(4) In Figure 5 and Supplementary Figure 6, please quantify the Oil-red O+ fat cells in the 2D and 3D lipogenic induction. Also in Fig. 6B, quantify the oil-red+MHC+ cells;

Thank you for this advice. We have quantified the oil-red O stained images in the result “Stimulate the fat deposition in chicken fibroblasts in 3D” using analysis software imageJ and the quantification of Oil-red O area was added to the corresponding graphs (Figure 5C, Figure S6C and S6F).

However, due to the unique structure of the 3D matrix, many MHC+ and Oil Red O+ double-positive cells overlap with each other across different Z-stack layers in 3D. This overlap makes it challenging to accurately position and quantify the double-positive cells as the different layers interfere with each other.

(5) In Figure 7, please show immunostaining images of collagen and other major ECMs;

Thank you for this question. We have tried to stain collagen networks the by the Picrosirius Red staining but failed. Instead, we employed the laminin immunostainings to confirm that the ECM contents in the 3D matrix is increasing steadily during cell culturation.

Please see Figure 7C. Lines 346-348.

“the laminin protein content was accumulated and increased steadily during 3D culturation (Figure 7C) “

(6) In Figure 8, please show hierarchical clustering analysis of whole transcriptomes of 3D_fibroblasts, 3D_MyoD, 3D+FI, and 3D_MyoD+FI. A Venn Diagram showing the overlap and distinct gene expression among these groups is also appreciated.

Thank you for the suggestion.

We added the hierarchical clustering analysis of whole transcriptomes of 3D_fibroblasts, 3D_MyoD, 3D+FI, and 3D_MyoD+FI using Euclidean distance with ward.D cluster method. Please see Figure 8B. The result showed that these groups formed two large clusters, in which the 3D+FI clustered separately and the 3D_fibroblasts, 3D_MyoD and 3D_MyoD+FI were more similar. Please see Figure 8B.

As the reviewer suggested, we also compared the transcriptomes of 3D_MyoD, 3D+FI, and 3D_MyoD+FI to the original 3D_fibroblasts to identify differentially expression genes (DEG) and then analyzed the overlap and distinct DEGs respectively. As shown in Figure 8D, the Venn Diagram showed that majority of DEG from 3D_MyoD+FI (3D_MyoD+FI versus 3D_fibroblasts) are overlapped with 3D_MyoD and 3D+FI, indicating that 3D_MyoD+FI are compatible with myogenic and adipogenic function.

Please see the revised Figure 8.

**Reviewer #2 (Recommendations For The Authors):**
In this study, the authors demonstrated a new approach for cultivated meat production using chicken fibroblasts. Specifically, the cells were cultured as 3D and induced muscle differentiation and lipid deposition. The manuscript contains a good set of data, which would be valuable to researchers in the fields of both cell-based meat and skeletal muscle biology. From the aspect of cultivated meat science, the rationale behind the idea is understandable, but it remains unclear whether the proposed approach was really the best choice to achieve their final goal. On the other hand, when we read this manuscript as a paper in skeletal muscle biology, the overall approach was not innovative enough and several uncertain issues remain. The authors should add more sufficient justifications, arguments, and discussions.(1) When considering their goal to produce edible meat products, the current approach has some concerns. First, there are issues with the approach used for the induction of myogenesis by MyoD transgene. This makes the end products GMO foods, which are not easily acceptable to a wide range of consumers. Next, the hydrogel was used for 3D tissue formation, but it is unclear whether this matrix type is edible, safe, and bio-comparable for cell-based meat production. The authors already discussed these points by excusing that the current work remains proof-of-concept. However, more careful considerations and justifications would be required.

Thank you for the suggestion.

We acknowledge that the current transgene myogenic induction method is not suitable for mass production of culture meat because of the GMO food concerns. We utilized the MyoD transgene as the means of myogenic transdifferentiation at the first place, because of the ease of genetic manipulation and maximum efficiency. We are current testing non-genomic integration tools such as chemical cocktails and modified RNAs for myogenic transdifferentiation.

When it comes to the applications of hydrogel in the food industry, certain types of hybrid hydrogels, such as those made from pectin or sodium polyacrylate, are not only edible but also safe for consumption. While GelMA hydrogel is typically utilized in tissue engineering and subsequent implantation in patients for therapeutic regenerative medicine purposes, it has not been commonly employed in food processing. In this study, we cultivated cells within GelMA hydrogel due to its durability and ease of use in cell culture. Moving forward, we plan to investigate alternative types of matrices to develop cultured meat suitable for food applications.

We have now described the GMO and hydrogel drawbacks in the discussion part. Please see lines 439-457.

“As a proof-of-concept, we utilized the transgene method to achieve maximum myogenic induction and the final products still retain the foreign transgene fragment in the cells’ genome. It is therefore posing a risk of genetic modified food which is not suitable for mass production. In the next step, other non-transgenic means such as non-integrating vectors, chemical reprogramming, modified RNAs, and recombinant transgene removal techniques will be explored to develop transgene-free end products. Another food safety concern in this study is the use of GelMA hydrogel for culture meat production. Due to its excellent biocompatibility and mechanical flexibility, GelMA-based hydrogel has demonstrated significant potential in scalable 3D cell culture for creating artificial tissue ranging in sizes from millimeters to centimeters. It is widely used in 3D cell culture and tissue engineering for regenerative medicine, but less common in food processing and agricultural applications. Due to its special photo-crosslinking properties, biocompatibility and degradability, it allows this material to be shaped into complex tissue structures by 3D printing or modelling. Many researchers have also used GelMA hydrogel as a scaffold for culture meat production (Jeong et al., 2022; Li et al., 2021; Park et al., 2023). Later research will carefully consider hydrogel as well as other types of scaffold biomaterials for cost-effective and food-safety compliant culture meat production (Bomkamp et al., 2022). ”

(2) From the view of skeletal muscle biology, the approaches (MyoD overexpression, hydrogel-based 3D tissue formation, and lipogenic induction) have already been tested.

Thank you for the insightful comments from the perspective of skeletal muscle cell biology. We totally agree that the current approaches including MyoD overexpression, 3D cell culture and lipogenic induction, were routine experiments in muscle cell biology. However, we want to highlight that utilization of these classical and robust muscle cell approaches, combine with the unique advantages of fibroblast cells (easily accessible, immortalized, cost-effective, ...) would provide a novel and practical avenue for culture meat production. We stated these issues in the revised manuscript in the discussion part.

Please see lines 511-515.

“In conclusion, we have effectively utilized immortalized chicken fibroblasts in conjunction with classical myogenic/adipogenic transdifferentiation approaches within 3D hydrogel to establish a cultured meat model. This model allows for the precise regulation of the synthesis of key components found in conventional meat, including muscle, fat, and ECM.”

(3) The common emphasis in this manuscript is to use the advantages of 3D culture for tissue differentiation. As the authors described, skeletal muscle is a highly aligned tissue. In this study, some results successfully demonstrated advantages in terms of myocyte alignment, maturation, and lipid deposition. However, the current results cannot address whether the entire 3D tissues maintained these advantageous characteristics or not. Because the method for 3D formation does not have any additional modifications to make the cells aligned, like micropatterning, scaffolding, or bioprinting.

Thank you for the suggestion.

We agree with the reviewer that the skeletal muscle tissues are composed of well organized, directional bundles of fibers, and the cell alignment would greatly affect the meat tenderness and sensory properties. Therefore, it is a desired attribute if the cells in the culture meat matrix could be aligned together. But this alignment would require sophisticated biomaterial engineering mainly involved in the scaffold manipulation which is beyond the scope of this study. The hydrogel used in this study formed different sizes of pores at random directions and we would expect the embedded cells to be totally non-directional. But we still found localized cell alignments in some parts of the gel matrix which confirming the cell-cell interactions, please see figure 3D. We describe this feature in the results part. In the future, we will be testing the application of physical or electrical stimulations to the matrix to see if we can align the cells better to make all the muscle cells in the whole matrix to align together.

Please see lines 186-190.

“The separate XY axis views of the orthogonal projections at different depths (Figure 3D) and a multi-angle video (Supplementary Video 2) also showed the several myotubes were aligned together. Nevertheless, many myotubes were oriented in different directions, preventing the entire matrix from aligning in one direction.”

(4) In the skeletal muscle, fat accumulation mainly occurs in adipocytes between myocytes. This means that "intra-" muscular fat deposition is identified. However, lipid deposition within myocytes also occurred in this preparation (Supplementary Figure 7C). This situation is not "intra-" muscular accumulation, which sounds different from what is going on in normal skeletal muscle tissues. Please explain what happened and what biological situations accounted for this. Also, the authors should clarify better how lipogenesis was induced in the 3D tissues, such as cell types (transdifferentiated myocytes, remained/un-transdifferentiated fibroblasts, or both).

Thank you for the very insightful question. We have revised the corresponding text to further explain the intramuscular fat distribution in different cell types in culture meat.

We totally agree with the reviewer that intramuscular fat accumulation may occur mainly in the intramuscular adipocytes. However, under some pathological and physiological conditions in human and animals, the lipid droplets were also abundantly observed inside myofibers (intramyocellular lipids within myofiber cytoplasm). For instance, high intramyocellular lipid content was found in insulin resistance patients and paradoxically in endurance trained athletes, (doi.org/10.1016/j.tem.2012.05.009), as well as in some farm animals under intensive selective breeding (doi:10.2174/1876142910901010059). In the current study, with the Oil Red O staining of lipid droplets, we identified lipid deposition in both the transdifferentiated myocytes and the remained un-transdifferentiated fibroblasts in the culture meat. This lipid distribution pattern is comparable to the intramuscular fat storage pattern observed in some human and animals, in which fat accumulation occurs in both myofibers (intramyocellular lipids) and intramuscular adipocyte cells (extramyocellular lipids) which reside within the muscle tissue bundle but between myofibers. We reason that current adipogenic induction treatment caused lipogenesis in both the MyoD-transdifferentiated cells and un-transdifferentiated fibroblasts. It is difficult to compare the absolute amount of lipids between these two types of cells via the Oil Red O staining. Also, it is almost impossible to separate these two types of cells from the 3D meat mimics. Thus, we can only confirm the lipid deposition occurs in both transdifferentiated myocytes and un-transdifferentiated fibroblasts, but without knowing which one is dominant and the major contributor to the intramuscular fat content in the culture meat.

Please see lines 486-492.

“In this study, the deposition of fat in the myotubes/myofibers facilitated the storage of significant lipid quantities in transdifferentiated muscle cells, known as intramyocellular lipids. Additionally, we observed Oil Red O staining in the remaining un-transdifferentiated fibroblasts, resembling cells of intramuscular adipocytes (extramyocellular lipids) found within muscle tissue. Hence, current adipogenic induction treatment caused lipogenesis in both the MyoD-transdifferentiated cells and un-transdifferentiated fibroblasts.”